# Wandering principal optical axes in van der Waals triclinic materials

Georgy A. Ermolaev[1], Kirill V. Voronin[2], Adilet N. Toksumakov[3], Dmitriy V. Grudinin[1], Ilia M. Fradkin[1], Arslan Mazitov[4], Aleksandr S. Slavich[3], Mikhail K. Tatmyshevskiy[3], Dmitry I. Yakubovsky[3], Valentin R. Solovey[1], Roman V. Kirtaev[1], Sergey M. Novikov[3], Elena S. Zhukova [3], Ivan Kruglov[1], Andrey A. Vyshnevyy[1], Denis G. Baranov [3], Davit A. Ghazaryan [3,5], Aleksey V. Arsenin [1,5], Luis Martin-Moreno [6,7], Valentyn S. Volkov[1,5] & Kostya S. Novoselov [8,9,10] ✉

Nature is abundant in material platforms with anisotropic permittivities arising from symmetry reduction that feature a variety of extraordinary optical effects. Principal optical axes are essential characteristics for these effects that define light-matter interaction. Their orientation – an orthogonal Cartesian basis that diagonalizes the permittivity tensor, is often assumed stationary. Here, we show that the low-symmetry triclinic crystalline structure of van der Waals rhenium disulfide and rhenium diselenide is characterized by wandering principal optical axes in the space-wavelength domain with above π/2 degree of rotation for in-plane components. In turn, this leads to wavelength-switchable propagation directions of their waveguide modes. The physical origin of wandering principal optical axes is explained using a multi-exciton phenomenological model and ab initio calculations. We envision that the wandering principal optical axes of the investigated low-symmetry triclinic van der Waals crystals offer a platform for unexplored anisotropic phenomena and nanophotonic applications.

Symmetry plays a pivotal role in fundamental laws of nature[1–7], including classical equations of motion, conservation laws, superposition principle, selection rules, and exchange interaction[8–11]. In condensed matter, it governs many of the material's mechanical, electronic, and optical properties, such as stress tensor, electron mobility, conductivity, refractive index, and allowed nonlinear processes, among others[12–15]. Highly symmetric atomic lattices, such as Al, Ni, and Au, result in isotropy of electronic and optical properties,

severely limiting their use[16–18]. For instance, they lack even-harmonic generation, birefringence, and chirality[4,19,20]. On the other hand, reducing the lattice's symmetry group leads to the emergence of anisotropy – the change of particular property in the observation direction[5,21]. The most known consequence is the birefringence phenomenon, wherein a birefringent material doubles an image[22]. This effect is just one of the numerous implications of anisotropic optical properties, traditionally described via the permittivity tensor[23].

[1]Emerging Technologies Research Center, XPANCEO, Dubai Investment Park First, Dubai, United Arab Emirates. [2]Donostia International Physics Center (DIPC), Donostia/San Sebastián 20018, Spain. [3]Moscow Center for Advanced Studies, Kulakova str. 20, Moscow 123592, Russia. [4]Institute of Materials, École Polytechnique Fédérale de Lausanne, 1015 Lausanne, Switzerland. [5]Laboratory of Advanced Functional Materials, Yerevan State University, Yerevan 0025, Armenia. [6]Instituto de Nanociencia y Materiales de Aragón (INMA), CSIC-Universidad de Zaragoza, 50009 Zaragoza, Spain. [7]Departamento de Física de la Materia Condensada, Universidad de Zaragoza, 50009 Zaragoza, Spain. [8]National Graphene Institute (NGI), University of Manchester, Manchester M13 9PL, UK. [9]Department of Materials Science and Engineering, National University of Singapore, Singapore 03-09, Singapore. [10]Institute for Functional Intelligent Materials, National University of Singapore, 117544 Singapore, Singapore. ✉e-mail: kostya@nus.edu.sg

It effectively describes the difference in refractive indices along various directions. This anisotropy produces complex isofrequency contours in the reciprocal space[24] enabling hyperbolic materials[21], ghost[1] and shear[10,25] polaritons, negative refraction[26,27], canalization of radiation[4], and many other intriguing wave phenomena.

Van der Waals (vdW) crystals offer a flexible and highly functional platform with a built-in anisotropy due to their fundamental difference between intralayer covalent and interlayer vdW bonding[28]. Therefore, such layered materials allow exotic light-matter interactions[29], resulting in exciton-[30], phonon-[31], edge-[32], and moiré-polaritons[33]. In most cases, this anisotropy is purely uniaxial, and the principal optical axes of the permittivity tensor are stationary with wavelength[28]. Some vdW crystals, however, have biaxial anisotropy because of the in-plane low-symmetry crystal structure[34–38]. Combined with non-orthogonally polarized in-plane exciton resonances[34], rhenium disulfide and rhenium diselenide can enable the wandering (wavelength-dispersive) direction of the principal optical axes of the permittivity tensor. Although the prediction of wavelength-dispersive principal optical axes dates back to 1928[39], experimental evidence of the discussed behavior has been elusive in inorganic crystals. We anticipate that more exotic optical responses and applications may be expected in materials with wandering principal optical axes, which can extend the evergrowing phenomena in low-symmetry nanophotonics[40].

In this work, we experimentally observed the rotation of principal optical axes in triclinic vdW crystals. We explained it via a bi-excitonic model, also recreating the wandering of such principal optical axes with first-principle calculations of the permittivity tensor. Here, only the individual components of the obtained permittivity tensor satisfy the Kramers–Kronig (KK) relations. In contrast, the generalized KK relation for crystallographic axes[41] is not applicable to triclinic rhenium disulfide (and diselenide). Hence, these crystals have extraordinary optical properties that set them apart from vdW and non-vdW crystals.

Furthermore, from a practical point of view, our near-field nanoimaging results reveal high wavelength sensitivity of light-matter interaction in triclinic vdW crystals, which can be leveraged for advanced light routing. Thus, triclinic van der Waals rhenium disulfide (and diselenide) offer a platform for anisotropic phenomena and next-generation nanophotonics.

## Results

### Impact of triclinic crystal structure on optical axes

$ReS_2$ and $ReSe_2$ are ideal materials for asymmetry-driven phenomena since they exhibit the lowest symmetry triclinic crystal structure[42], shown in Fig. 1a–c. Consequently, they received considerable interest in recent works[34–37,43–45], which reported a high linear and nonlinear optical anisotropy originating from non-collinear excitons[34]. In particular, the angle between the polarizations of excitons[46] is about 70° instead of the expected 90°. It arises from Peierls' distortion of the 1 T structure (Fig. 1a)[36]. This feature should, naturally, cause nontrivial optical responses, such as non-orthogonal self-hybridized polaritons[47]. Therefore, a more thorough investigation of the anisotropic dielectric tensor $\hat{\varepsilon}$ of $ReS_2$ and $ReSe_2$ remains a significant challenge both because their dielectric tensors cannot be diagonalized in Cartesian coordinates[39] and for their great demand for low-symmetry photonics.

Nevertheless, according to Onsager's theorem[48], their dielectric tensors are symmetric ($\hat{\varepsilon} = \hat{\varepsilon}^T$). They thus can be divided into Hermitian (Re[$\hat{\varepsilon}$]) and skew-Hermitian (Im[$\hat{\varepsilon}$]) parts (Fig. 1d–e), primarily responsible for polarization and losses, respectively. It is worth noting that the diagonalization basis for Hermitian and skew-Hermitian tensors can differ and vary with wavelengths, as schematically illustrated in Fig. 1d–e, which can result in wavelength-dispersive principal optical axes. In fact, principal optical axes rotation explains the effects observed in earlier reports[35,38,49] on optical properties of $ReS_2$ (Supplementary Note 1).

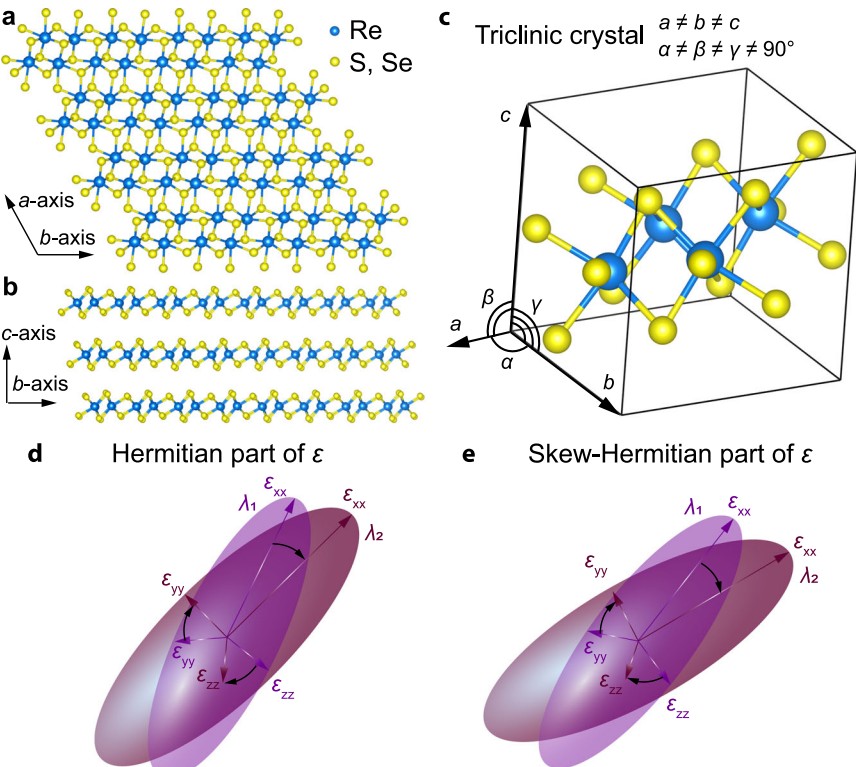

**Fig. 1 | Conceptualization of wavelength-dispersive principal optical axes in triclinic crystals.** Crystal structure of $ReS_2$ and $ReSe_2$ (**a**) along the $c$-axis and (**b**) along the $a$-axis, (**c**) three-dimensional view of the unit cell, where $\alpha$, $\beta$, and $\gamma$ are crystallographic angles of triclinic crystal. Schematic illustration of wandering principal optical axes for (**d**) Hermitian and (**e**) skew-Hermitian parts of dielectric tensors. $\varepsilon_{xx}$, $\varepsilon_{yy}$, and $\varepsilon_{zz}$ stands for dielectric permittivities in the basis of principal optical axes along principal optical axes for two wavelengths $\lambda_1$ and $\lambda_2$.

## Physical origins of wandering principal optical axes

To visualize this effect, we prepared ReS$_2$ and ReSe$_2$ samples (Fig. 2a, b and Supplementary Note 2) and measured polarised transmittance (Fig. 2c) around the exciton resonances. Figure 2c demonstrates how the angle for maximum transmittance shifts for different excitons, showing that the principal optical axes change with exciton resonances. In order to capture its wavelength dependence, we provide polarization spectra in Fig. 2d–e for ReS$_2$ and in Supplementary Note 2 for ReSe$_2$. Note that excitonic spectral dips vanish at certain polarizations (Fig. 2d), indicating the orientation of excitons. Of immediate interest are wandering (wavelength-dispersive) principal optical axes, shown in Fig. 2e and Supplementary Note 2. In fact, a recent study[50] showed that the principal optical axis at 550 and 650 nm tilts by 3° and 2°, respectively, with respect to the $b$-axis for few-layer ReS$_2$, which is close to our 7° and 8° observed for bulk ReS$_2$ (see Fig. 2e). At large wavelengths, the principal optical axes almost coincide with the crystallographic axes (Fig. 2e). However, the principal optical axes vary rapidly at fundamental exciton frequencies and then demonstrate complex behavior for high-energy photons, owing to the material's rich excitonic structure[51]. Still, the crystallographic axes influence the

position of the principal optical axes since, at the fundamental exciton resonances, the principal optical axes switch from the crystallographic $b$-axis to the $a$-axis (Fig. 2e). At infrared wavelengths, this wandering of principal optical axes reaches 65° whereas, for the whole spectral range, it exceeds 110° change, as seen in Fig. 2e.

Furthermore, this extraordinary optical response influences the Raman spectra (Supplementary Note 3). For instance, polarisation-resolved Raman measurements reveal the change of phonon modes' preferential direction when the excitation wavelength switches from 532 nm to 633 nm and then to 780 nm (Supplementary Note 3). Although phonon modes' directions have more complex behavior since they depend not only on the orientation of the principal optical axes but on the phonon modes themself, their dispersion follows a similar pattern to principal optical axes (Supplementary Note 3). This trend is unique to ReS$_2$ and ReSe$_2$, as we demonstrate in Supplementary Note 4, exemplifying a highly anisotropic As$_2$S$_3$ with static principal optical axes[52]. Indeed, As$_2$S$_3$ also has a reduced symmetry, which in principle, may cause a similar effect of wandering principal optical axes. However, unlike ReS$_2$ and ReSe$_2$, the crystal structure of As$_2$S$_3$ is close to orthorhombic phase with the following crystallographic parameters[52]:

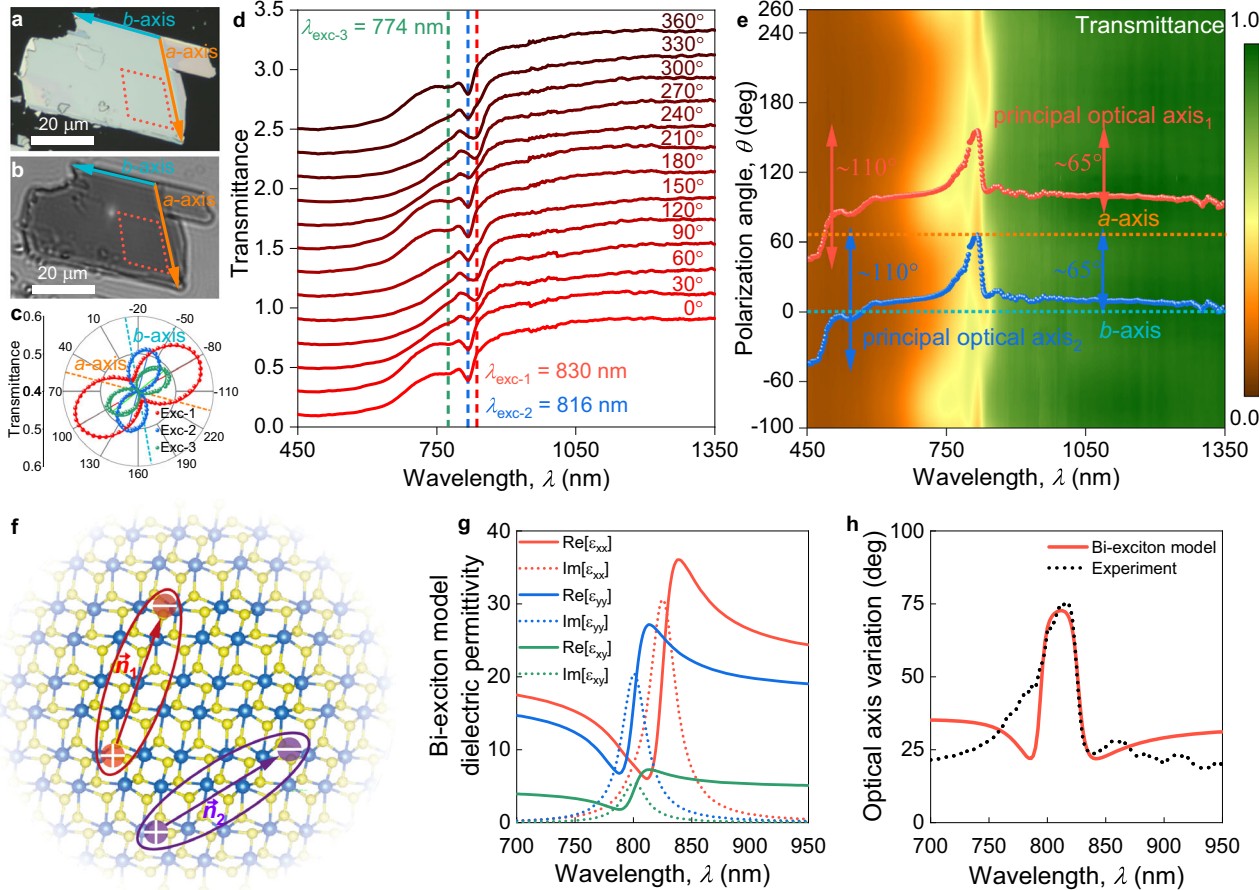

**Fig. 2 | Observation of wandering (wavelength-dispersive) principal optical axes in triclinic ReS$_2$.** (**a**) Optical and (**b**) ellipsometry micrographs of bulk ReS$_2$. Red dashed lines show the region for polarized microtransmittance measurements. (**c**) Polarized transmittance of bulk ReS$_2$ presented in panel (**a**), for three different exciton wavelengths of 830 nm (exc-1), 816 nm (exc-2), and 774 nm (exc-3). Each curves shifted by 0.2 for clarity. Polarized transmittance (**d**) spectra and (**e**) heatmap. In panel (**d**) dashed lines show the positions of fundamental excitons of bulk ReS$_2$. In panel (**e**), red and blue points show the positions of in-plane principal optical axes. Dashed lines correspond to the crystallographic $a$-axis (orange line) and $b$-axis (cyan line). Zero degree corresponds to the crystallographic $b$-axis. The red and blue points are obtained through the fitting of polarization-resolved

microtransmittance at each wavelength (see Methods section Determination of principal optical axes). Arrows show the maximum position change of principal optical axes. (**f**) Depiction of non-orthogonal excitons (phenomenological theory). Solid lines represent the binding between electron and hole in exciton. Arrows shows the preferential direction of excitons and $\boldsymbol{n_1}$ and $\boldsymbol{n_2}$ are unit vectors describing the in-plane polarization of the corresponding excitonic transition. (**g**) Dielectric tensor corresponding to the bi-exciton model. Solid lines show the real parts of dielectric permittivity, while dashed lines show the imaginary parts of dielectric permittivity. (**h**) Principal optical axes orientation as a function of wavelength. Solid red lines show principal optical axis change predicted by bi-exciton model. Dashed line is experimental positions of principal optical axis.

$a = 0.42546(4)$ nm, $b = 0.95775(10)$ nm, $c = 1.14148(10)$ nm, $\alpha = 90°$, $\beta = 90.442°$, and $\gamma = 90°$; because the monoclinic angle $\beta$ differs from 90° by just 0.442(4)°. Moreover, $As_2S_3$ is transparent in the measured spectral interval (450–1350 nm) implying that its excitons lie below 450 nm, and hence, their effect is negligible[52]. In other words, an illustration of static principal optical axes in $As_2S_3$ highlights the nontrivial behavior of wandering principal optical axes in $ReS_2$ and $ReSe_2$ since the observed effect requires both strongly reduced crystal symmetry and the presence of material's directional resonances: in our case excitons.

The observed behavior of the principal optical axes of $ReS_2$ and $ReSe_2$ can be described with a phenomenological bi-exciton model (Fig. 2f and Supplementary Note 5). According to this model, the permittivity tensor of $ReS_2$ in the visible range can be expressed as:

$$\hat{\varepsilon}(\omega) = \hat{\mathbb{I}}\varepsilon_\infty + f_1 \frac{\omega_1^2}{\omega_1^2 - \omega^2 - i\omega\gamma_1}\boldsymbol{n_1}\otimes\boldsymbol{n_1^*} + f_1\frac{\omega_1^2}{\omega_1^2 - \omega^2 - i\omega\gamma_1}\boldsymbol{n_2}\otimes\boldsymbol{n_2^*} \quad (1)$$

where $\omega_{1,2}$ is the resonant frequency of the exciton resonance, $\gamma_{1,2}$ is its non-radiative decay rate, $f_{1,2}$ is the rescaled oscillator strength, and $\boldsymbol{n_{1,2}} = (n_x, n_y, n_z)^T$ is a unit vector describing the in-plane polarization of the corresponding excitonic transition. By varying the parameters of the permittivity model ($\omega_i, \gamma_i, f_i$), we managed to find a dielectric tensor (Fig. 2g), which reproduces the wandering effect of $ReS_2$ principal optical axes (Fig. 2h) within a phenomenological bi-exciton model. The satisfactory agreement between the two-exciton model and the experiment corroborates the leading role of excitons in the observed behavior.

### Real-space nanoimaging of wandering principal optical axes

Wandering of $ReS_2$ and $ReSe_2$ principal optical axes opens the door to wavelength-switchable optics for efficient light manipulation. As a practical demonstration, we show the effect of wavelength–dispersive principal optical axes on waveguide mode propagation direction using a scattering scanning near-field optical microscopy (s-SNOM) in the transmission scheme, depicted in Fig. 3a. This scheme has no angular rotation (Supplementary Note 6) which makes it advantageous over the reflection scheme. Notably, the principal optical axes vary rapidly at fundamental exciton frequencies (see Fig. 2e and Supplementary Note 2). Therefore, for measurements, we focused on $ReSe_2$ because it provides a strong variation in the orientation of the principal optical axes within the measured wavelength range of our s-SNOM setup (Methods). To eliminate the edge effect on the near-field image when

launching the waveguide modes and launch those modes isotropically, we created a circular hole (the inset in Fig. 3a) inside the $ReSe_2$ sample. It allows us to visualize the asymmetry of waveguide modes (Fig. 3b–d) caused by material anisotropy only: Fig. 3b–d show elliptical light propagation. As anticipated, these ellipses rotate with wavelength change, as seen from the position of their major axes in Fig. 3e–g (theoretical background of direction change which is provided in Supplementary Notes 7–9). Notably, the observed near-field mode is an interference between the air and waveguide modes. Still, according to our analysis, the air mode's contribution to the rotation of the mode's propagation direction is negligible with respect to the wavelength (see Supplementary Note 9). Hence, wandering (wavelength-dispersive) principal optical axes offer a platform to manipulate light without additional structuring and engineering.

### Anisotropic dielectric tensors with wandering optical axes

Given the strong wavelength dispersion of the principal optical axes, it is challenging to describe the optical responses of $ReS_2$ and $ReSe_2$ correctly. Hence, we fitted the polarized transmittance spectra within the isotropic approximation as the initial step (see Supplementary Note 10). This approach yields a refractive index of about 4 in the infrared range, close to earlier reports[35,38,49], and allows for distinguishing the fundamental excitonic transitions. In the next step, we irradiated samples with unpolarized light to obtain the optical properties averaged over polarization angles. Notably, the resulting optical constants do not follow Kramers–Kronig relations (see Supplementary Note 10) in contrast to other anisotropic vdW materials[53]. Consequently, $ReS_2$ and $ReSe_2$ exhibit anomalous optical responses even for unpolarized light due to the wavelength-dispersive principal optical axes.

To better understand the wandering of principal optical axes, we performed first-principle calculations of monolayer, bilayer, trilayer, and bulk $ReS_2$ and $ReSe_2$ anisotropic permittivity tensors, shown in Fig. 4 and Supplementary Note 9. As expected, the off-diagonal tensor elements (Fig. 4a and g) are nonzero, and tensors are not diagonalizable on the orthogonal basis (Supplementary Note 11) because of the crystal's low symmetry. Nonetheless, we can decouple Hermitian and skew-Hermitian parts of tensors and diagonalize them separately, as shown in Fig. 4 and Supplementary Note 11. The diagonalization process also gives a diagonalization basis, which, in the case of the dielectric tensors, coincides with principal optical axes. Moreover, it allows us to directly observe a dramatic change of principal optical axes orientations from theoretical calculations (Fig. 4c–f and i-l), which

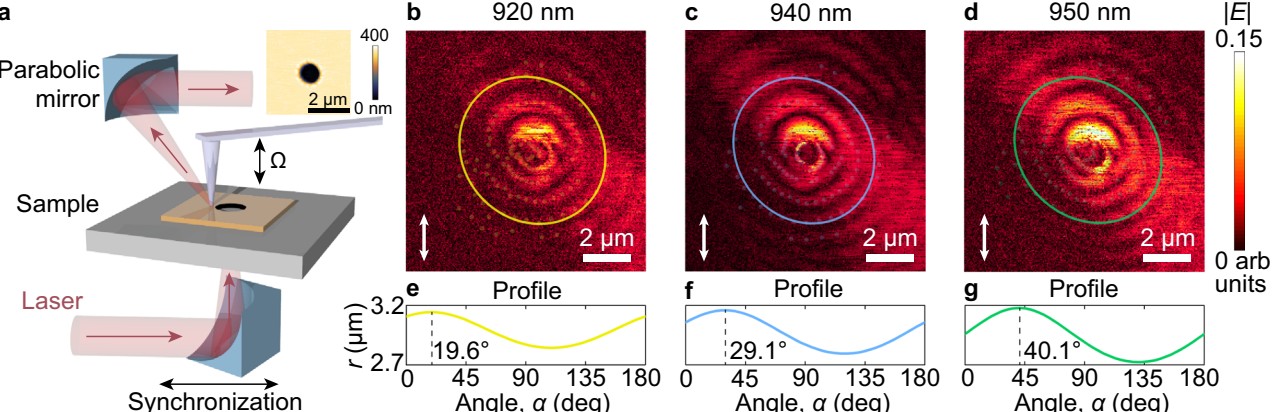

**Fig. 3 | Real-space nanoimaging of wandering (wavelength-dispersive) principal optical axes in triclinic $ReSe_2$. (a)** Sketch of the experimental configuration for the near-field measurements in the transmission mode. The inset is a height micrograph of the hole patterned in the $ReSe_2$ sample. $\Omega$ is an oscillation frequency of a near-field microscope cantilever. Near-field micrographs of waveguide mode at wavelengths of (**b**) 920 nm, (**c**) 940 nm, and (**d**) 950 nm. Ellipses are guides to an eye of mode propagation. Arrows indicate the incident light polarization. Dependence of the length of the radius vector of the ellipse on the angle between the radius vector and the incident polarization for (**e**) 920 nm, (**f**) 940 nm, and (**g**) 950 nm. The dotted line marks the angles between the ellipse's major axis and the incident polarization. The comparison of measured near-field with the calculated near-field within three-exciton model is provided in Supplementary Note 8.

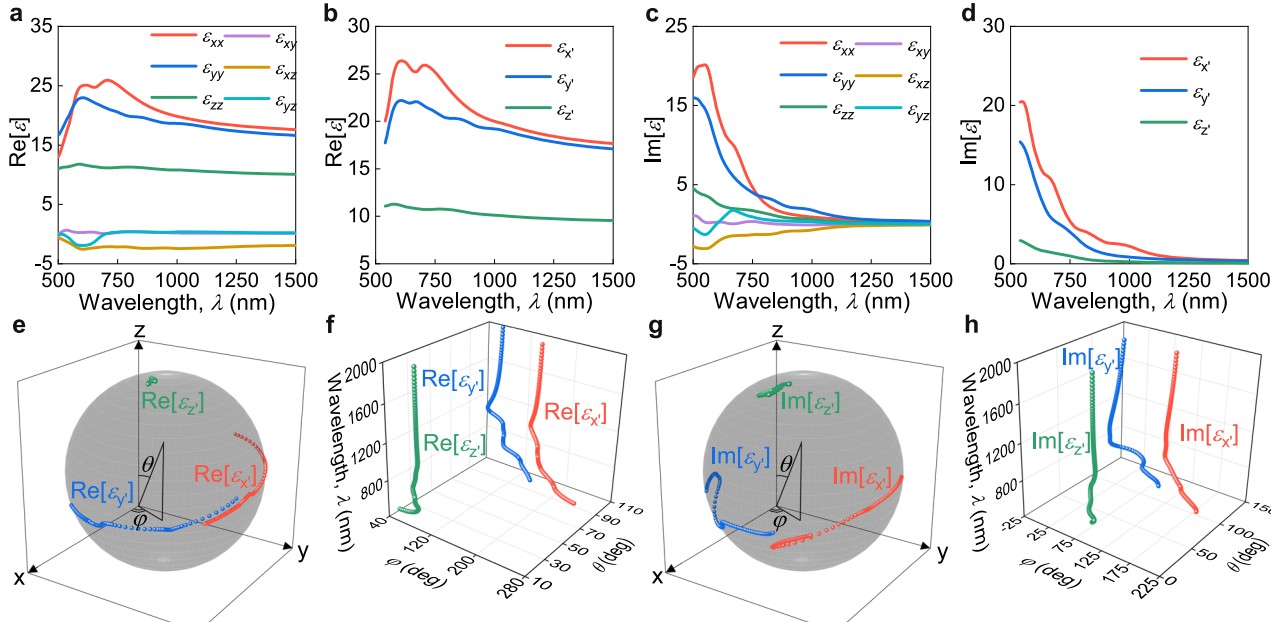

**Fig. 4 | First-principle calculations of bulk ReS₂ dielectric tensor.** (**a**) Hermitian part of the dielectric tensor. (**b**) Hermitian components of the dielectric tensor after the diagonalization process. (**c**) Skew-Hermitian part of the dielectric tensor. (**d**) Skew-Hermitian components of the dielectric tensor after the diagonalization process. (**e**) Three-dimensional view of principal optical axes variation for the Hermitian part of the dielectric tensor. Axes are dimensionless and serve as a reference for eyes. Grey sphere is also a guideline for eyes. (**f**) Wavelength dependence of principal optical axes positions for the Hermitian part of the dielectric tensor in polar coordinates ($\varphi, \theta$ in panel (**e**)). (**g**) Three-dimensional view of principal optical axes variation for the skew-Hermitian part of the dielectric tensor. Axes are dimensionless and serve as a reference for eyes. Grey sphere is also a guideline for eyes. (**h**) Wavelength dependence of principal optical axes positions for the skew-Hermitian part of the dielectric tensor in polar coordinates ($\varphi, \theta$ in panel (**g**)).

agree with the experimental findings in Fig. 2, thereby unambiguously verifying the effect of wandering (wavelength-dispersive) principal optical axes in ReS₂ and ReSe₂. Moreover, the non-straight orientation of the principal optical axes of the permittivity tensor leads to slanted isofrequency surfaces with respect to the global z-axis (Supplementary Figure 21), which may enable interesting transmission phenomena, such as negative refraction and the super-prism effect[54]. Hence, the unique dielectric tensors of ReS₂ and ReSe₂ (Fig. 2 and Supplementary Figure 17) provide great flexibility in optical engineering.

## Discussion

The permittivity tensor is the key optical characteristic of any artificial or natural material. It describes the material's polarizability via the permittivity values and fundamental directions called principal optical axes, where birefringence is absent. Although for an overwhelming majority of inorganic materials, dielectric constant values are wavelength-dispersive, enabling numerous phenomena such as ultraslow light and Fano resonances[55,56], principal optical axes remain static, which limits nanophotonics since this "degree of freedom" is unavailable. In this regard, triclinic van der Waals materials offer a platform for the emergence of wandering (wavelength-dispersive) principal optical axes appearing in far- and near-fields and in quantum mechanical calculations. This unconventional optical response was demonstrated for rhenium disulfide (diselenide) and was shown to originate from non-orthogonal exciton resonances. Furthermore, the properties associated with wandering principal optical axes can be observed in fields other than optics by considering the material's non-Hermiticity arising from broken crystal symmetries. We also anticipate wandering principal optical axes in other low-symmetry crystals with triclinic and monoclinic structures, including GeS₂[57], Lu₂SiO₅[58], CdWO₄[25], β-phase Ga₂O₃[10], and many others[10,25,57,58]. These materials offer interesting opportunities for wavelength-switchable metamaterials, metasurfaces, waveguides, and cavities[59–62].

## Methods

### Sample preparation

Bulk ReS₂ and ReSe₂ crystals were purchased from 2D Semiconductors (Scottsdale, USA) and micromechanically cleaved down on top of required substrates (Si/SiO₂ and glass). Those substrates were subsequently decontaminated in acetone, isopropanol alcohol, and deionized water before the cleavage and then subjected to oxygen plasma removing the ambient adsorbates. Following plasma treatment, substrates were subjected to thermal treatment at temperatures of 120 °C and then exposed to scotch-tape from Nitto Denko Corporation (Osaka, Japan) with loaded bulk crystals of ReS₂ and ReSe₂. Eventually, the scotch-tape was removed, completing the cleavage procedure. The thickness of as-papered thin ReS₂ and ReSe₂ crystals was measured by an atomic force microscope (NT-MDT Spectrum Instruments, Ntegra II) in HybriD Mode using HA_NC tips with resonant frequency of 140 kHz and spring constant of 3.5 N/m.

### Determination of principal optical axes

We used polarized microtransmittance measurement technique implemented on our Accurion nanofilm_ep4 ellipsometer to determine the principal optical axes. During the measurements, we aligned the polarizer and analyzer of the ellipsometer and fitted the obtained polarized microtransmittance for each wavelength by the expression: $T(\theta, \lambda) = a^2\cos^4(\theta - \varphi) + b^2\sin^4(\theta - \varphi) + 2ab\cos^2(\theta - \varphi)\sin^2(\theta - \varphi)\cos(\Delta\phi)$, where $T(\theta, \lambda)$ is the polarized microtransmittance, which depends on the polarizer's/analyzer's angle $\theta$, and the incident wavelength $\lambda$. $a^2$ and $b^2$ are the transmittances of beams polarized along in-plane principal optical axes, $\Delta\phi$ is a phase difference between transmitted rays polarized along principal optical axes, and $\varphi$ indicates the angular position of the principal optical axis (see blue points in Fig. 2e), whereas another principal optical axis is given by the sum $\varphi + 90°$ (see red points in Fig. 2e).

## Scanning near-field optical microscopy

The fabricated hole in ReSe$_2$ was characterized by the amplitude- and phase-resolved scattering-type scanning near-field optical microscopy (s-SNOM) measurements using the "NeaSNOM" setup (Neaspec GmbH). The s-SNOM works as an atomic force microscope (AFM) in a tapping mode with a Pt-coated silicon tip oscillating at the resonance frequency of $\Omega \approx 280$ kHz with an amplitude of $\sim 50$ nm. In the s-SNOM working at transmission configuration, the ReSe$_2$ hole is illuminated from below by a linearly polarized light at a normal angle to the sample surface focused by a bottom parabolic mirror. As a light source, we used Ti:Sapphire continuous wave tuning laser (TiC, AVESTA Lasers and Optical Systems) with fiber coupling output, working at a wavelength range of $\lambda = 700-1000$ nm. While mapping the near-field signal and AFM topography around the hole with a scan area of $10 \times 10$ μm$^2$, the illumination from the bottom parabolic mirror always remained aligned with the hole due to its synchronization moving with the sample during the scan. A top parabolic mirror collects the tip-scattered near-field signal and directs it into the highly sensitive photodetector. To achieve a clear near-field image, the optical background was suppressed by demodulation of the detected signal at high-order harmonic frequency $n\Omega$ ($n = 2, 3, 4$) and using an interferometric pseudoheterodyne detection scheme with a modulated reference beam via oscillating mirror. In this work, the demodulation signal at the third harmonic ($3\Omega$) was taken, which is enough for background-free near-field detection.

## First-principle calculations

Optical constants of the ReS$_2$ and ReSe$_2$ crystals were calculated within density functional theory (DFT) and GW approximation, as implemented in VASP package[63]. First, the atomic positions of both crystals were relaxed until the interatomic forces decreased below $10^{-3}$ eV/Å, while their unit cells were fixed. The lattice parameters were $a = 6.378$ Å, $b = 6.417$ Å, $c = 6.461$ Å with $\alpha = 91.62°$, $\beta = 119.07°$, $\gamma = 105.115°$ for ReS$_2$ and $a = 6.716$ Å, $b = 6.602$ Å, $c = 6.728$ Å with $\alpha = 104.90°$, $\beta = 91.82°$, $\gamma = 118.94°$ for ReSe$_2$. Next, we obtained ground-state one-electron wavefunctions from DFT and used them to initialize the GW routines. Finally, we calculated the imaginary and real parts of the frequency-dependent dielectric function within GW approximation and derived the refractive indices and extinction coefficients of the material. The cutoff energy for the plane-wave basis was set to 500 eV, while the first Brillouin zone was sampled with a $\Gamma$-centred $6 \times 6 \times 6$ grid. The exchange correlation effects were described with a generalized gradient approximation (Perdew-Burke-Ernzerhof functional), and the behavior of wavefunctions in the core region was reconstructed with the projector augmented wave pseudopotentials.

## Data availability

The relevant raw and generated data supporting the key findings of this study are available in the figshare database under accession code https://figshare.com/s/a1edc12b21d3a36315ab (https://doi.org/10.6084/m9.figshare.24967593). All data are available from the corresponding author upon a request.

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

## Acknowledgements

K.S.N. acknowledges support from the Ministry of Education, Singapore (Research Centre of Excellence award to the Institute for Functional Intelligent Materials, I-FIM, project No. EDUNC-33-18-279-V12) and from the Royal Society (UK, grant number RSRP\R\190000). S.M.N. acknowledges the financial support from the Ministry of Science and Higher Education (agreement No. 075-15-2022-1150). A.S.S. and A.N.T. gratefully acknowledge the financial support from the RSF (grant No. 22-19-00558). D.A.G., A.V.A., and V.S.V. acknowledge support by the Higher Education and Science Committee of the Ministry of Education, Science, Culture, and Sport of the Republic of Armenia Project No. 23RL-2A031. L.M.M acknowledges Project PID2020-115221GB-C41, financed by MCIN/AEI/10.13039/501100011033, and the Aragon Government through Project Q-MAD.

## Author contributions

D.G.B., D.A.G., A.V.A., L. M.-M., K.S.N., and V.S.V. suggested and directed the project. G.A.E., D.V.G., A.S.S., M.K.T., D.I.Y., V.R.S., S.M.N., and E.S.Z. performed the measurements and analyzed the data. A.N.T., R.V.K., and D.A.G. prepared the samples. G.A.E., K.V.V., D.V.G., I.M.F., A.M., I.K., A.A.V., and D.G.B. provided theoretical support. G.A.E. wrote the original manuscript. G.A.E., D.G.B., D.A.G., A.V.A., L. M.-M., K.S.N., and V.S.V. reviewed and edited the paper. All authors contributed to the discussions and commented on the paper.

## Competing interests

The authors declare no competing interests.
