## [Peer Review File · Nature Communications]

Wandering principal optical axes in van der Waals triclinic materialsREVIEWER COMMENTS

Reviewer #1 (Remarks to the Author):

In this work, the authors investigate the peculiar effect of principal optical axes moving with the wavelength in triclinic 2D materials ReS₂ and ReSe₂. This is an important finding that should attract the interest from the researchers working on 2D materials as well as the general audience in optics. The work seems to have been carried out with sufficient care, and the analyses are scientifically sound. There are a few points that the authors should consider to strengthen the impact of the paper.

1. The phenomenon of the optical axes of ReS₂ deviating from the crystallographic axes has been reported before [ACS Nano 16, 9222]. In this paper, a deviation of 2.4 degree was measured for the wavelength of 650 nm. Since the current authors measured for the entire range of visible spectrum, the current work clearly has novelty. On the other hand, the authors should compare their results against the previously reported one.
2. In line 111, numbers such as 65 degrees and 110 degrees appear. However, I cannot read these numbers off Figure 2e. On a related matter, it is not clear how the red and blue data points in Figure 2e were obtained.
3. In Figure 2d, I think the angles indicated are measured with respect to the optical image (Fig. 2a) and have no scientific meaning. I think it would be better to convert the angles with respect to the b-axis direction.
4. Polarized Raman spectroscopy of ReS₂ and ReSe₂ with different excitation wavelengths has been reported before but was not properly analyzed [Nanoscale Horizons 5, 308 (2020)]. With the help of current results, I think the authors may be able to explain the previously reported results.
5. Some of the text in the figures are too small.

Reviewer #2 (Remarks to the Author):

Georgy A. Ermolaev et al. reported the wavelength-dispersive principal optical axes in triclinic ReS₂ with both experimental observation and theoretical calculation. This phenomenon was attributed to non-orthogonally polarized in-plane exciton resonances. Based on this phenomenon, the author further demonstrated wavelength-switchable propagation directions of the waveguide modes on scattering-SNOM. The reported phenomenon is intriguing, I would like to recommend it if authors can well address the following questions.

1. Figure 2b, the axis label is not clear;
2. Experimental details on the polarized transmission measurement (Figure 2c) should be provided, such as direction of the incident polarization respect to the sample orientation, and with both polarizer and analyzer?
3. Please explain why the wandering principle optical axes is unique to ReS₂ while not observed in As₂S₃? The exciton information of As₂S₃ should be mentioned for easier understanding for general readers.
4. Excitonic effect and bandgap of ReS₂ may change with its thickness, I am wondering how the wandering principle optical axes depends on thickness (monolayer, 2L, 3L)?
5. The influence of principal optical axes rotation on Raman spectra of phonon modes is very complicated, with both electronic resonance effect (Phys. Rev. Lett. 2013, 111 (12), 126801.) and dispersive dielectric constants (Science Bulletin 2020, 65 (22), 1894-1900.). The specific vibration pattern of phonon mode may also affect the electron-phonon coupling and the polarization-dependent Raman intensity. 785 nm excitation is close to the exciton frequency in ReS₂, therefore the electronic resonance should be also considered in the related discussion.
6. Since the principal optical axes vary rapidly at fundamental exciton frequencies, why does the author choose the excitation from 920-950 nm instead of the exciton frequencies in s-SNOM experiment (Figure 3)?

Reviewer #3 (Remarks to the Author):

In the report “Wandering principal optical axes in van der Waals triclinic materials” G. A. Ermolaev et al., discuss photonic effects in ReSe₂ and ReS₂ crystals. Overall, I believe that the report is novel and interesting. However, I believe that some of the technical points need further justification and support. Revisions are suggested to address the points below.

1. The linewidth of resonances in ReS₂ near 800 nm appears to be extremely narrow. Is the linewidth consistent with other reports on ReS₂? Are these results consistent with Fig.S9? Have the authors taken nano-optical measurements in the spectral range around 800 nm where negative permittivity components are seen in Fig. 2g?
2. The authors go to some length to discuss flaws in other analyses in their supplementary materials. Yet, support for the optical constants provided in their own report is not entirely clear. For example, the transfer matrix calculations in Fig. S2b are shown to be inconsistent with the data in Fig.S2a, but similar transfer matrix calculations performed with the permittivity the authors claim in Fig.S6a are not even shown. Comparison between claimed optical constants and the measured wavevectors in the SNOM measurements is similarly lacking, but should be readily possible.
3. How are the authors confident that the modes visualized in Fig.3 are waveguide modes within ReSe₂ and not air modes (see eg. Ref [34]) for a discussion of waveguide modes in ReS₂ and air modes in SNOM experiments)?
4. The authors write that they visualize “asymmetry of the waveguide modes (Figures 3b-d) caused by

material anisotropy only.” Yet, waveguide modes with weak confinement are known to have angular dependence in SNOM measurements (see eg. Ref. [29]). The authors are suggested to extract the SNOM wavevectors from their data as a function of the in-plane angle, perform established angular correction protocols (described, for instance in Ref. [34]) and compare the results with calculations using their reported optical constants.

5. I found the opening sentence of the abstract “Nature is abundant in material systems...” confusing. The authors are suggested to rephrase.

6. The authors cite ref [26] for negative refraction of polaritons on pg. 2. However, while Ref. [26] demonstrated so-called ‘bending-free’ refraction, negative refraction was not claimed in this report. Negative refraction of polaritons has been reported in H. Hu et al., Science 379, 558 (2023), and A. J. Sternbach et al., Science 379, 555 (2023).

7. The authors mention a “super-prism effect” on page 6 but no reference is provided. The authors are suggested to include a reference.

**SUMMARY OF THE CHANGES INTRODUCED IN MANUSCRIPT NCOMMS-
23-26804-T**

**"WANDERING PRINCIPAL OPTICAL AXES IN VAN DER WAALS
TRICLINIC MATERIALS"**

We provide below the detailed responses to the Reviewers' concerns and comments (with the marked changes in green).

RESPONSES TO REVIEWER #1's COMMENTS AND CONCERNS:

GENERAL ASSESSMENT: In this work, the authors investigate the peculiar effect of principal optical axes moving with the wavelength in triclinic 2D materials ReS₂ and ReSe₂. This is an important finding that should attract the interest from the researchers working on 2D materials as well as the general audience in optics. The work seems to have been carried out with sufficient care, and the analyses are scientifically sound. There are a few points that the authors should consider to strengthen the impact of the paper.

RESPONSE: The authors thank the Reviewer for his kind assessment of the article's findings and the motivating report. All of the concerns raised by the Reviewer have been carefully considered and responded to point-by-point. The resulting changes are marked in green in the revised version of our manuscript.

COMMENT 1: The phenomenon of the optical axes of ReS₂ deviating from the crystallographic axes has been reported before [ACS Nano 16, 9222]. In this paper, a deviation of 2.4 degree was measured for the wavelength of 650 nm. Since the current authors measured for the entire range of visible spectrum, the current work clearly has novelty. On the other hand, the authors should compare their results against the previously reported one.

RESPONSE: We thank the Reviewer for drawing our attention to this relevant publication [J.M. Park et al., "Precise determination of offset between optical axis and Re-chain direction in rhenium disulfide", ACS Nano, 16(6), 9222-9227 (2022)], where the authors identify ~3° and 2° offset between the principal optical axis and crystallographic *b*-axis for few-layer ReS₂ for 550 nm and 650 nm, respectively. In the case of our bulk (150 nm-thick) ReS₂ sample, we discovered the effect

of wandering of the principal optical axis with respect to wavelength (see Figure 2e of the main text) covering the whole spectral range from 450 nm to 1350 nm. Notably, for the wavelengths mentioned (550 nm and 650 nm), we obtained similar $\sim 8^\circ$ and 7° tilts between the principal optical axis and crystallographic *b*-axis. Hence, the work [J.M. Park et al., “Precise determination of offset between optical axis and Re-chain direction in rhenium disulfide”, ACS Nano, 16(6), 9222-9227 (2022)] supports our findings. Accordingly, we included the comparison with the results of the publication [J.M. Park et al., “Precise determination of offset between optical axis and Re-chain direction in rhenium disulfide”, ACS Nano, 16(6), 9222-9227 (2022)] in the revised version of our manuscript:

Of immediate interest are wandering (wavelength-dispersive) principal optical axes, shown in Figure 2e and Supplementary Section 2. In fact, a recent study⁵⁰ showed that the principal optical axis at 550 and 650 nm tilts by 3° and 2° , correspondingly, with respect to the *b*-axis for few-layer ReS₂, which is close to our 7° and 8° observed for bulk ReS₂ (see Figure 2e).

[50] Park, J. M., Lee, S., Na, W., Kim, K. & Cheong, H. Precise Determination of Offset between Optical Axis and Re-Chain Direction in Rhenium Disulfide. ACS Nano 16, 9222–9227 (2022).

COMMENT 2: In line 111, numbers such as 65 degrees and 110 degrees appear. However, I cannot read these numbers off Figure 2e. On a related matter, it is not clear how the red and blue data points in Figure 2e were obtained.

RESPONSE: We are grateful to the Reviewer for the comment and are sorry for the technical issues regarding the quality loss of the presented Figures. We have fixed this issue in the revised version of our manuscript and included a clear indication of 65° and 110° degree changes of principal optical axes in Figure 2e. Meanwhile, the red and blue data points (indicating the positions of the principal optical axes) are obtained through the fitting of polarized microtransmittance $T(\theta, \lambda)$ for each wavelength using the formula $T(\theta, \lambda) = a^2 \cos^4(\theta - \varphi) + b^2 \sin^4(\theta - \varphi) + 2ab \cos^2(\theta - \varphi) \sin^2(\theta - \varphi) \cos(\Delta\varphi)$, which depends on the polarizer/analyzer angle θ , and the incident wavelength λ . Here, a^2 and b^2 are the transmittances of beams polarized along in-plane principal optical axes, $\Delta\varphi$ is a phase difference between rays polarized along principal optical axes, and φ indicates the angular position of the principal optical axis (see blue points in Figure 2e), whereas the other principal optical axis is given by the sum $\varphi+90^\circ$ (see red points in Figure 2e). Therefore, we accordingly modified the corresponding chapter and Figure 2e in the revised version of our manuscript

Still, the crystallographic axes influence the position of the principal optical axes since, at the fundamental exciton resonances, the principal optical axes switch from the crystallographic *b*-axis to the *a*-axis (Figure 2e). At infrared wavelengths, this wandering of principal optical axes reaches 65° whereas, for the whole spectral range, it exceeds 110° change, as seen in Figure 2e.

Determination of principal optical axes. We used the polarized microtransmittance measurement technique implemented on our Accurion nanofilm_ep4 ellipsometer to determine the principal optical

axes. During the measurements, we aligned the polarizer and analyzer of the ellipsometer to obtain polarized microtransmittance for each wavelength using the following formula: $T(\theta, \lambda) = a^2 \cos^4(\theta - \varphi) + b^2 \sin^4(\theta - \varphi) + 2ab \cos^2(\theta - \varphi) \sin^2(\theta - \varphi) \cos(\Delta\phi)$, where $T(\theta, \lambda)$ is the polarized microtransmittance, which depends on the polarizer's/analyzer's angle θ , and the incident wavelength λ . a^2 and b^2 are the transmittances of beams polarized along in-plane principal optical axes, $\Delta\phi$ is a phase difference between rays polarized along principal optical axes, and φ indicates the angular position of the principal optical axis (see blue points in Figure 2e), whereas another principal optical axis is given by the sum $\varphi + 90^\circ$ (see red points in Figure 2e).

Revised version of Figure 2:

Figure 2 | Observation of wandering (wavelength-dispersive) principal optical axes in triclinic ReS_2 . **a**, Optical and **b**, ellipsometry micrographs of bulk ReS_2 . **c**, Polarized transmittance of bulk ReS_2 presented in panel (a), for three different excitons at 830 nm (exc-1), 816 nm (exc-2), and 774 nm (exc-3). Polarized transmittance **d**, spectra and **e**, contour map. Red and blue points show the positions of in-plane principal optical axes. Dashed lines correspond to the crystallographic a -axis (red line) and b -axis (blue line). **Zero degree corresponds to the crystallographic b -axis**. **f**, Depiction of non-orthogonal excitons (phenomenological theory). **g**, Dielectric tensor used for the bi -exciton model. **h**, Principal optical axes dependence on wavelength. **The red and blue points are obtained through the fitting of microtransmittance spectra for each wavelength (see Methods section Determination of principal optical axes)**.

COMMENT 3: In Figure 2d, I think the angles indicated are measured with respect to the optical image (Fig. 2a) and have no scientific meaning. I think it would be better to convert the angles with respect to the *b*-axis direction.

RESPONSE: We thank the Reviewer for pointing this out. Following the Reviewer's suggestion, we converted the angles with respect to the crystallographic *b*-axis direction (see the revised version of Figure 2 in the response to the previous Comment).

COMMENT 4: Polarized Raman spectroscopy of ReS₂ and ReSe₂ with different excitation wavelengths has been reported before but was not properly analyzed [Nanoscale Horizons 5, 308 (2020)]. With the help of current results, I think the authors may be able to explain the previously reported results.

RESPONSE: We are grateful to the Reviewer for the question. Indeed, in [Nanoscale Horizons, 5, 308 (2020)], the authors observed a similar Raman wavelength-dependent response of polarized Raman modes of maximum intensity with respect to the *b*-axis. They did not suggest the physical reasons for this observation, while the discovered effect of wandering principal optical axes in our study explains this Raman response analysis easily. Briefly, principal optical axes determine the light-matter interaction efficiency for a given polarization direction, which influences the Raman signal's intensity, and results in the shift of the Raman mode's maximum intensity. A similar explanation is also valid for our results. Hence, we included the suggested reference [Nanoscale Horizons, 5, 308 (2020)] in the Raman section of Supplementary Information to give a better explanation of the wandering principal optical axes phenomenon:

The comparison between the wavelength-dispersive change of both maximum intensity of phonon modes and principal optical axes rotation reveals a similar trend, as seen in Supplementary Figure 4g. Briefly, principal optical axes determine the light-matter interaction efficiency for a given polarization direction, which influences the Raman signal's intensity and results in the shift of the Raman mode's maximum intensity. The same explanation is also valid for previous Raman results, for example, in³, the authors report a similar Raman wavelength-dependent response of polarized modes of maximum intensity with respect to the *b*-axis.

Supplementary Figure 4: Polarized Raman measurements for ReS₂. Characteristic Raman spectra for **a**, 532 nm, **b**, 633 nm, and **c**, 780 nm. Polarized phonon mode intensities of III and V phonon modes for **d**, 532 nm, **e**, 633 nm, and **f**, 780 nm. **g**, The comparison of principal optical axes rotation with phonon mode position of maximum intensity.

COMMENT 5: Some of the text in the figures are too small.

RESPONSE: We thank the Reviewer for pointing this out and are sorry for the poor compilation of the Figures. To address this issue we increased the font size for all small text through the Figures.

RESPONSES TO REVIEWER #2's COMMENTS AND CONCERNS:

GENERAL ASSESSMENT: Georgy A. Ermolaev et al. reported the wavelength-dispersive principal optical axes in triclinic ReS₂ with both experimental observation and theoretical calculation. This phenomenon was attributed to non-orthogonally polarized in-plane exciton resonances. Based on this phenomenon, the author further demonstrated wavelength-switchable propagation directions of the waveguide modes on scattering-SNOM. The reported phenomenon is intriguing, I would like to recommend it if authors can well address the following questions.

RESPONSE: We are grateful to the Reviewer for the recognition of the importance of our work. In the following sections, we present our responses to each of the comments/suggestions raised by the Reviewer. The resulting changes are marked in green in the revised version of our manuscript.

COMMENT 1: Figure 2b, the axis label is not clear

RESPONSE: We thank the Reviewer for pointing this out and are sorry for the poor compilation of the label of Figure 2b axis. To address this issue we increased the fonts of the axis label in Figure 2b.

COMMENT 2: Experimental details on the polarized transmission measurement (Figure 2c) should be provided, such as direction of the incident polarization with respect to the sample orientation, and with both polarizer and analyzer?

RESPONSE: We thank the Reviewer for the comment and are sorry for such an omission. In the methods section of the revised version of our manuscript, we included experimental details on polarized microtransmittance measurements and updated the corresponding starting point (zero degree) with respect to crystallographic b -axis in Figures 2c and e to clearly note the orientations of polarizer and analyzer with respect to the sample:

Determination of principal optical axes. We used the polarized microtransmittance measurement technique implemented on our Accurion nanofilm_ep4 ellipsometer to determine the principal optical axes. During the measurements, we aligned the polarizer and analyzer of the ellipsometer to obtain polarized microtransmittance for each wavelength using the following formula: $T(\theta, \lambda) = a^2 \cos^4(\theta - \varphi) + b^2 \sin^4(\theta - \varphi) + 2ab \cos^2(\theta - \varphi) \sin^2(\theta - \varphi) \cos(\Delta\phi)$, where $T(\theta, \lambda)$ is the polarized microtransmittance, which depends on the polarizer's/analyzer's angle θ , and the incident wavelength λ . a^2 and b^2 are the transmittances of beams polarized along in-plane principal optical axes, $\Delta\phi$ is a phase difference between rays polarized along principal optical axes, and φ indicates the angular position of the principal optical axis (see blue points in Figure 2e), whereas another principal optical axis is given by the sum $\varphi + 90^\circ$ (see red points in Figure 2e).

COMMENT 3: Please explain why the wandering principle optical axes is unique to ReS₂ while not observed in As₂S₃? The exciton information of As₂S₃ should be mentioned for easier understanding for general readers.

RESPONSE: We thank the Reviewer for the comment. Indeed, As₂S₃ also has a reduced symmetry, which in principle, may cause a similar effect of wandering principal optical axes. However, unlike ReS₂ and ReSe₂, the crystal structure of As₂S₃ is close to orthorhombic phase with the following crystallographic parameters: $a = 0.42546(4)$ nm, $b = 0.95775(10)$ nm, $c = 1.14148(10)$ nm, $\alpha = 90^\circ$, $\beta = 90.442^\circ$, and $\gamma = 90^\circ$, because the monoclinic angle β differs from 90° by just $0.442(4)^\circ$. Moreover, As₂S₃ is transparent in the measured spectral interval (450 – 1350 nm) implying that its excitons lie below 450 nm, and hence, their effect is negligible. In other words, an illustration of static principal optical axes in As₂S₃ demonstrates the nontrivial behavior of wandering principal optical axes in ReS₂ and ReSe₂ since the observed effect requires both strongly reduced crystal symmetry and the presence of material's directional resonances: in our case excitons. To make a stronger emphasis, we present our recent study of the optical response of As₂S₃ at Arxiv's database (<https://arxiv.org/abs/2309.01989>). As a result, we included the following explanation in the revised version of the manuscript:

This trend is unique to ReS₂ and ReSe₂, as we demonstrate in Supplementary Section 4, employing a highly anisotropic As₂S₃ with static principal optical axes⁵². Indeed, As₂S₃ also has a reduced symmetry, which in principle, may cause a similar effect of wandering principal optical axes. However, unlike ReS₂

and ReSe_2 , the crystal structure of As_2S_3 is close to orthorhombic phase with the following crystallographic parameters⁵²: $a = 0.42546(4)$ nm, $b = 0.95775(10)$ nm, $c = 1.14148(10)$ nm, $\alpha = 90^\circ$, $\beta = 90.442^\circ$, and $\gamma = 90^\circ$; because the monoclinic angle β differs from 90° by just $0.442(4)^\circ$. Moreover, As_2S_3 is transparent in the measured spectral interval (450 – 1350 nm) implying that its excitons lie below 450 nm, and hence, their effect is negligible⁵¹. In other words, an illustration of static principal optical axes in As_2S_3 demonstrates the nontrivial behavior of wandering principal optical axes in ReS_2 and ReSe_2 since the observed effect requires both strongly reduced crystal symmetry and the presence of material's directional resonances: in our case excitons.

COMMENT 4: Excitonic effect and bandgap of ReS_2 may change with its thickness. I am wondering how the wandering principle optical axes depend on thickness (monolayer, 2L, 3L)?

RESPONSE: We are grateful to the Reviewer for the interesting question about wandering principal optical axes for atomically thin layers (monolayer, bilayer, trilayer, *etc.*). We agree with the Reviewer that excitonic effects and bandgap in ReS_2 change with its thickness [for example, see M. Gehlmann et al., “Direct observation of the band gap transition in atomically thin ReS_2 ”, *Nano Lett.*, 17(9), 5187 - 5192 (2017)]. However, the recent work [J. Wu, et al., “Monolayer behavior in bulk ReS_2 due to electronic and vibrational decoupling”, *Nat. Comm.*, 5, 3252 (2014)] demonstrates that ReS_2 has one of the weakest dependencies of optical and electronic properties on the number of atomic layers among vdW materials owing to the Peierls distortion of its 1T crystal structure, which prevents ordered stacking and minimizes the interlayer overlap of wavefunctions. As a result, we expect that the behavior/extent of wandering principal optical axes for monolayer, bilayer, and trilayer ReS_2 will be similar to its bulk. Additionally, we provided first-principle calculations of anisotropic permittivity tensors of 1L, 2L, and 3L ReS_2 , which show a similar optical response as bulk ReS_2 in Figure 4 of the main text. Consequently, we modified the main text of the revised version of our manuscript by the following statement:

To better understand the phenomenon of wandering principal optical axes, we performed first-principle calculations of monolayer, bilayer, trilayer, and bulk ReS_2 and ReSe_2 anisotropic permittivity tensors, shown in Figure 4 and Supplementary Section 11.

Also, we modified the revised version of our Supplementary Information by:

Moreover, the excitonic effects and the size of the bandgap of ReS_2 may change with its thickness, especially for monolayer, bilayer, and trilayer cases¹¹. However, another recent work¹¹ demonstrates that ReS_2 has one of the smallest dependencies of optical and electronic properties on the number of atomic layers among vdW materials owing to Peierls distortion of its 1T crystal structure, which prevents ordered stacking and minimizes the interlayer overlap of wavefunctions. As a result, we expect that the observed behavior/extent of the wandering principal optical axes for monolayer, bilayer, and trilayer ReS_2 will be similar to its bulk. Additionally, we present first-principle calculations of anisotropic permittivity tensor of monolayer (see Supplementary Figure 18), bilayer (see

Supplementary Figure 19), and trilayer (see Supplementary Figure 20) ReS_2 , that show optical response similar to bulk ReS_2 presented in Figure 4 of the main text.

Supplementary Figure 18: First-principle calculations of dielectric tensor of monolayer ReS_2 . **a**, Hermitian part of the dielectric tensor. **b**, Hermitian components of the dielectric tensor after the diagonalization. **c**, Skew-Hermitian part of the dielectric tensor. **d**, Skew-Hermitian components of the dielectric tensor after the diagonalization. **e**, Three-dimensional view of principal optical axes variation for the Hermitian part of the dielectric tensor. **f**, Wavelength dependence of principal optical axes positions for the Hermitian part of the dielectric tensor in polar coordinates (see ϕ , θ in panel (e)). **g**, Three-dimensional view of principal optical axes variation for the skew-Hermitian part of the dielectric tensor. **h**, Wavelength dependence of principal optical axes positions for the skew-Hermitian part of the dielectric tensor in polar coordinates (see ϕ , θ in panel (g)).

Supplementary Figure 19: First-principle calculations of dielectric tensor of bilayer ReS_2 . **a**, Hermitian part of the dielectric tensor. **b**, Hermitian components of the dielectric tensor after the diagonalization. **c**, Skew-Hermitian part of the dielectric tensor. **d**, Skew-Hermitian components of the dielectric tensor after the diagonalization. **e**, Three-dimensional view of principal optical axes variation for the Hermitian part of the dielectric tensor. **f**, Wavelength dependence of principal optical axes positions for the Hermitian part of the dielectric tensor in polar coordinates (see ϕ , θ in panel (e)). **g**, Three-dimensional view of principal optical axes variation for the skew-Hermitian part of the dielectric tensor. **h**, Wavelength dependence of principal optical axes positions for the skew-Hermitian part of the dielectric tensor in polar coordinates (see ϕ , θ in panel (g)).

Supplementary Figure 20: First-principle calculations of dielectric tensor of trilayer ReS_2 . **a**, Hermitian part of the dielectric tensor. **b**, Hermitian components of the dielectric tensor after the diagonalization. **c**, Skew-Hermitian part of the dielectric tensor. **d**, Skew-Hermitian components of the dielectric tensor after the diagonalization. **e**, Three-dimensional view of principal optical axes variation for the Hermitian part of the dielectric tensor. **f**, Wavelength dependence of principal optical axes positions for the Hermitian part of the dielectric tensor in polar coordinates (see φ , θ in panel (e)). **g**, Three-dimensional view of principal optical axes variation for the skew-Hermitian part of the dielectric tensor. **h**, Wavelength dependence of principal optical axes positions for the skew-Hermitian part of the dielectric tensor in polar coordinates (see φ , θ in panel (g)).

COMMENT 5: The influence of principal optical axes rotation on Raman spectra of phonon modes is very complicated, with both electronic resonance effect (Phys. Rev. Lett. 2013, 111 (12), 126801.) and dispersive dielectric constants (Science Bulletin 2020, 65 (22), 1894-1900.). The specific vibration pattern of phonon mode may also affect the electron-phonon coupling and the polarization-dependent Raman intensity. 785 nm excitation is close to the exciton frequency in ReS_2 , therefore the electronic resonance should be also considered in the related discussion.

RESPONSE: We thank the Reviewer for the comment and agree that both electronic resonance effects [Phys. Rev. Lett., 111, 126801 (2013)] and dispersive dielectric constants [Science Bulletin, 65, 1894 - 1900 (2020)] can influence the Raman spectra, especially, for 785 nm excitation wavelength, which is near the fundamental exciton's energy in ReS_2 . Nonetheless, both effects cannot explain the wavelength-dependent shift in the maximum intensity of Raman modes with respect to the b -axis, while the discovered phenomenon of wandering principal optical axes in our study, explains this Raman effect easily. Briefly, principal optical axes determine the light-matter interaction efficiency for a given polarization direction, which influences the Raman signal's intensity and results in the shift of the Raman mode's maximum intensity. Though we agree that our description is qualitative, it already gives an insight into the influence of the wandering principal optical axes on the Raman signal and explains previous intriguing results, for example, see [Nanoscale Horizons, 5, 308 (2020)]. Since the detailed Raman analysis should involve, at least, three non-trivial effects (wandering principal optical axes, electronic resonance effect, and dispersive dielectric constants), it certainly deserves its own publication. Hence, we

provided only a qualitative picture in Supplementary Information and commented that a comprehensive Raman investigation should also take the electronic resonance effect, and dispersive dielectric constants into account to fully describe the Raman signal:

The comparison between the wavelength-dispersive change of both maximum intensity of phonon modes and principal optical axes rotation reveals a similar trend, as seen in Supplementary Figure 4g. Briefly, principal optical axes determine the light-matter interaction efficiency for a given polarization direction, which influences the Raman signal's intensity and results in the shift of the Raman mode's maximum intensity. The same explanation is also valid for previous Raman results, for example, in³, the authors report a similar Raman wavelength-dependent response of polarized modes of maximum intensity with respect to the *b*-axis.

Though our description is qualitative, it already gives an insight into the influence of wandering principal optical axes on the Raman signal and explains previous intriguing results³. Notably, the detailed Raman analysis should involve, at least, three non-trivial effects, such as wandering principal optical axes, electronic resonance effect, and dispersive dielectric constants. Our qualitative results are summarized in Supplementary Figure 4g.

Supplementary Figure 4: Polarized Raman measurements for ReS₂. Characteristic Raman spectra for **a**, 532 nm, **b**, 633 nm, and **c**, 780 nm. Polarized phonon mode intensities of III and V phonon modes for **d**, 532 nm, **e**, 633 nm, and **f**, 780 nm. **g**, The comparison of principal optical axes rotation with phonon mode position of maximum intensity.

COMMENT 6: Since the principal optical axes vary rapidly at fundamental exciton frequencies, why does the author choose the excitation from 920-950 nm instead of the exciton frequencies in s-SNOM experiment (Figure 3)?

RESPONSE: We thank the Reviewer for the question. Indeed, principal optical axes vary rapidly at fundamental exciton frequencies. This is exactly why we chose the range of 920 – 950 nm for ReSe₂. These wavelengths correspond to the fundamental exciton frequencies as presented in

Supplementary Figures 2 and 3. The reason for choosing ReSe₂ instead of ReS₂ for near-field measurements is technical: the fundamental excitons of ReSe₂ have larger wavelengths, which greatly simplifies near-field measurements since the large wavelengths allow working with a large diffraction limit, which increases the allowable error when adjusting parabolic mirrors that focus radiation at the probe-sample interface and collect scattered near-field data. In our measurement setup, the minimum step of movement of parabolic mirrors is about 80 nm, which greatly complicates the obtaining of a pure near-field signal (filtered from the background) in the visible range. Another reason for the difficulty with 780 nm is that the laser of the near-field microscope, which measures the amplitude of the probe's oscillations, operates exactly in this range. Even taking into account the optical filter we have built into the system, the illumination of the sample from below near this range can lead to incorrect detection of the amplitude of the probe's oscillations, which in turn will lead to incorrect measurements at best, and at worst will damage the probe itself (see Figure 3a for the schematics of our measurement setup).

Supplementary Figure 3: Observation of wavelength-dispersive principal optical axes in triclinic ReSe₂. **a**, Optical image of exfoliated bulk ReSe₂. **b**, Polarized transmittance map. The red and blue points show the position of the principal optical axes.

Since the comment is of great importance, we included it in the main text:

As a practical demonstration, we show the effect of wavelength–dispersive principal optical axes on waveguide mode propagation direction using a scattering scanning near-field optical microscopy (s-SNOM) in the transmittance scheme, depicted in Figure 3a, since the reflectance scheme has angular rotation (Supplementary Section 6). Notably, the principal optical axes vary rapidly at fundamental exciton frequencies (see Figure 2e and Supplementary Section 2). Therefore, for measurements, we focused on ReSe₂ because it provides a strong variation in the orientation of the principal optical axes within the measured wavelength range of our s-SNOM setup (Methods).

RESPONSES TO REVIEWER #3's COMMENTS AND CONCERNS:

GENERAL ASSESSMENT: In the report “Wandering principal optical axes in van der Waals triclinic materials” G. A. Ermolaev et al., discuss photonic effects in ReSe₂ and ReS₂ crystals. Overall, I believe that the report is novel and interesting. However, I believe that some of the technical points need further justification and support. Revisions are suggested to address the points below.

RESPONSE: We are grateful to the Reviewer for the thorough examination and the balanced and positive assessment of our work. Below we address point-by-point all the comments and questions raised by the Reviewer after careful consideration.

COMMENT 1: The linewidth of resonances in ReS₂ near 800 nm appears to be extremely narrow. Is the linewidth consistent with other reports on ReS₂? Are these results consistent with Fig.S9? Have the authors taken nano-optical measurements in the spectral range around 800 nm where negative permittivity components are seen in Fig. 2g?

RESPONSE: We thank the Reviewer for the comment and agree that the linewidth of resonances presented in Figure 2g around 800 nm is quite narrow for our ReS₂. This is because our model accounts only for non-orthogonal *bi-excitonic phenomenological* dielectric permittivity. It was developed to explain the physical background behind the wandering of the principal optical axis, not its quantitative behavior as can be seen from Figure 2h. Nevertheless, we agree with the Reviewer that the axis title “Dielectric permittivity” is misleading, therefore we replaced it with “Phenomenological dielectric permittivity” in the revised version of our manuscript:

COMMENT 2: How are the authors confident that the modes visualized in Fig.3 are waveguide modes within ReSe₂ and not air modes (see eg. Ref [34]) for a discussion of waveguide modes in ReS₂ and air modes in SNOM experiments)?

RESPONSE: We would like to express our gratitude to the Reviewer for this valuable comment. Accordingly, after a deep analysis of our SNOM results (see Supplementary Information Sections

7, 8, and 9), we came to the conclusion that the observed near-field mode of ReSe₂, which is presented in Figure 3, is a hybrid of the waveguide and air modes. Once again, we would like to thank the Reviewer for drawing our attention to this point. We modified our manuscript as follows:

As anticipated, these ellipses rotate with wavelength change, as seen from the position of their major axes in Figure 3e-g (see the theoretical background of direction change which is provided in Supplementary Sections 7-9). **Notably, the observed near-field mode is an interference between the air and waveguide modes. Still, according to our analysis, the air mode's contribution to the rotation of the mode with respect to the wavelength is negligible (see Supplementary Section 9).** Hence, wandering (wavelength-dispersive) principal optical axes offer an unprecedented platform to manipulate light without additional structuring and engineering.

In the revised version of Supplementary Information, we added:

Supplementary Section 8: Theoretical background of waveguide mode's direction change – near-field calculations analysis.

We consider a 150-nm-thick ReSe₂ layer on a glass substrate in our numerical near-field calculations. The refractive index of the glass is 1.4, whereas the permittivity of ReSe₂ is described by the phenomenological 3-exciton model (see Supplementary Section 5 and Supplementary Figure 6 for details). We illuminate a 500-nm-radius hole in the layer by a plane wave normally incident from the substrate and study the z-component (perpendicular to the structure surface) of the electric field, $|E_z|$, in the air at a distance of 5 nm from the air/ReSe₂ interface. Due to a peculiar structure of the 3-exciton model's permittivity tensor ($\hat{\epsilon}_{xz} = \hat{\epsilon}_{yz} = 0$), the normally incident plane wave does not induce a z-component in the background field, $E_{bg}^z = 0$. Therefore, the z component of the total electric field equals to the corresponding component of the scattered field, $E_{tot}^z = E_{sc}^z$.

Supplementary Figure 8 presents a series of near-field distributions, categorized by the incident light wavelengths (columns) and angles of the flake's rotation (rows). The three columns represent wavelengths of 920 nm, 940 nm, and 950 nm, while the seven rows display fields for rotation angles of $\{0^\circ, 30^\circ, 60^\circ, 90^\circ, 120^\circ, 150^\circ, 180^\circ\}$. Both 0° and 180° rotations yield identical results and are shown for the ease of comparison with adjacent panels. The incident light polarization (indicated by white arrows) is vertical in all panels.

Upon comparing the columns, we observe that the slight rotation of the bright (yellow in the image) electrostatic-dipole-like-polarized field in the immediate vicinity of the hole is most apparent in the first and last rows. The waves surrounding the hole also exhibit slight shifts. Concurrently, the flake's rotation has a relatively weak effect on the bright, dipole-like field, which remains predominantly vertically polarized regardless of the flake's orientation. However, the waves surrounding the hole clearly rotate along with the flake. This supports the notion that the observed waves are associated with the waveguide modes of the anisotropic ReSe₂ layer.

Supplementary Figure 8: Near-field distribution near a circular hole in the ReSe₂ layer. Three columns correspond to 920 nm, 940 nm, and 950 nm wavelengths. Eight rows correspond to six linear polarizations of incident light taken with a 30-degree step, left-handed (LCP), and right-handed (RCP) circular polarization. Near field distribution pattern in the considered anisotropic structure mainly depends on the polarization of the incident beam, but also, slightly varies with wavelength.

The numerical calculations not only provide us with the absolute value of the electric field but also, its phase (see Supplementary Figure 9). To analyze them together, we employ a straightforward data processing algorithm, illustrated for the case of a non-rotated flake and a wavelength of 920 nm in Supplementary Figure 9. The algorithm consists of the following steps: *(i)* transformation of coordinates from (x, y) to (r, φ) (see Supplementary Figure 9); *(ii)* referencing to^{1,9}, and application of the Parzen window (see the window shape in Supplementary Figure 10) to smoothen the computational domain's edge and the hotspot near the hole (see Supplementary Figure 9); and finally, *(iii)* performing a Fourier transform on the obtained complex field for the radial coordinate and analyzing its absolute value (see Supplementary Figure 9). The application of the Parzen window enables us to extend the Fourier integral to infinite limits and avoid parasitic oscillations caused by the Gibbs phenomenon.

Supplementary Figure 9: Processing of the near-field distribution for a circular hole in the ReSe₂ layer. **a**, The absolute value and **b**, phase of the near-field distribution are shown for vertical polarization of 920 nm light incident on a non-rotated flake. **c-d**, The same fields after transformation to polar coordinates. **e-f**, Multiplication of the obtained field with a real-valued Parzen window does not affect the phase but smooths the absolute value of the field near the hole and edges of the computational domain. **g**, The absolute value of the obtained field Fourier image. The white dashed line indicates the wavevector of air modes, and the observed bright lines indicate the dispersion of air and waveguide modes.

Supplementary Figure 10: The shape of the Parzen filter applied to the near-field distributions.

The procedure described above allows us to effectively distinguish the contributions of different modes to the overall field. As depicted in Supplementary Figure 9, all the modes are situated in the upper half-space, indicating positive values of the radial wavevector, k_r . This confirms the evident fact that the modes propagate outward from the hole. One of the branches follows the horizontal white dashed line, which corresponds to the light wavevector in the air. This confirms that this particular mode is associated with freely propagating air modes scattered out by the hole. Moreover, the angle dependence of this mode's amplitude exhibits a typical vertical dipole radiation pattern – with dips at $\varphi_{deeps} = \{0^\circ, 180^\circ\}$ and peaks in between. However, the peaks are significantly shifted from the expected positions of $\pm 90^\circ$, as the actual polarization currents induced by the hole differ significantly from a point dipole that is vertically aligned with the polarization of the incident light. In addition to the air modes, we also observe branches of waveguide modes of two types. The first type exhibits strong angular dispersion and propagates in almost all possible directions, while the second type is observed in a narrow range of angles near $\varphi = \{0^\circ, 180^\circ\}$ and possesses much larger wavevectors.

Based on our findings regarding the mode structure, we conclude that the waves observed in Supplementary Figure 8 (as well as in Supplementary Figure 9) are not associated with either air modes or waveguide modes individually, but rather with their interference: $|E_z(\varphi, r)| \propto |E_{air}(\varphi)e^{ik_0r} + E_{waveguide}(\varphi)e^{ik_{wg}r}|$. Therefore, the radial period of these waves is inversely proportional to the difference between their wavevectors, given by $\lambda_{waves} = \frac{2\pi}{k_{wg} - k_0} = \frac{\lambda_{wg}\lambda_0}{\lambda_0 - \lambda_{wg}} = \frac{\lambda_0}{\beta_{wg} - 1}$, where β_{wg} represents the index of the waveguide mode and λ_{waves} , λ_{wg} , λ_{wg} correspond to the period of the waves in the near-field, wavelengths of the waveguide, and air modes, respectively.

By following the described procedure, we obtain the Fourier spectra (refer to Supplementary Figure 11) for various wavelengths and angles of flake rotation. All the observed effects are consistent with what we have already observed in real space (see Supplementary Figure 8), but they are much more straightforward. Notably, the transformation of the angular dispersion of the waveguide modes with wavelength becomes more apparent. The rotation of the flake results in a corresponding rotation of the waveguide modes, while the angular position of the "dipole-like" air mode remains mostly unaffected.

Supplementary Figure 11: Fourier spectra of the near-field distribution for a circular hole in the ReSe₂ layer. Each panel demonstrates the absolute value of the Fourier transform of the filtered near field, $|F[E_x(\varphi, r) \cdot f_{\text{parzen}}(r)](\varphi, k_r)|$. The

three columns correspond to wavelengths of 920, 940, and 950 nm, while the even rows correspond to flake rotations of 0°, 30°, 60°, 90°, 120°, 150°, and 180°. The gray axes indicate the orientation of the flake, and the white vertical arrows show the polarization of the incident light. The white dashed horizontal lines correspond to the wavevector of air modes $k_0 = \pm \frac{2\pi}{\lambda_0}$.

COMMENT 3: The authors go to some length to discuss flaws in other analyses in their supplementary materials. Yet, support for the optical constants provided in their own report is not entirely clear. For example, the transfer matrix calculations in Fig. S2b are shown to be inconsistent with the data in Fig.S2a, but similar transfer matrix calculations performed with the permittivity the authors claim in Fig.S6a are not even shown. Comparison between claimed optical constants and the measured wavevectors in the SNOM measurements is similarly lacking, but should be readily possible.

RESPONSE: We thank the Reviewer for the comment and made the required changes to support our optical response description. The mentioned Section in Supplementary Information presents a thorough examination of previous works on ReS₂ and ReSe₂ meant to shed light on a fundamental difference between static and wandering principal optical axes. Nevertheless, we agree with the Reviewer that it is worth providing not only transmittance from static principal optical axes in comparison with experimental values, but also, transmittance from wandering principal optical axes. As a result, we updated our Supplementary Figure 2. Clearly, wandering principal optical axes have a good match with the experiment compared to static principal optical axes, which fail to reproduce the measured transmittance spectra even qualitatively. Nonetheless, one can notice a slight difference between the values of experimental and calculated transmittance spectra. We attribute this to a slight difference in absolute values of actual and evaluated dielectric permittivities (from first-principle calculations). Meanwhile, our data gives almost a perfect match for the effect of wandering principal optical axes, especially in the vicinity of 750 nm, where the rotation of principal optical axes has a maximum.

As fairly noticed by the Reviewer, the direct comparison between measured and calculated wavevectors is a complex task because the observed mode is a hybrid of waveguide and air modes. Nevertheless, to address the Reviewer's comment, we compare the experimental and calculated Fourier transformations (see Supplementary Figure 13), which qualitatively coincide with each other.

In accordance, we modified the revised version of our manuscript as follows:

Supplementary Section 1: Peculiarity of optical response of ReS₂.

Materials with rotating optical axes have more degrees of freedom for optical response than those with fixed principal optical axes. As a result, their optical performance cannot be fully described by a dielectric function with static principal axes. At the same time, it might be possible to describe a particular optical measurement by an effective dielectric tensor having fixed principal axes. However, the obtained effective dielectric tensor might look unphysical or fail to describe other measurements.

For example, work¹ extracts optical properties from near-field measurements (see Supplementary Figure 1a). From the Kramers–Kronig relations, one would expect the peak of optical absorption to appear for ϵ_y (see Supplementary figure 1b). We encounter a similar Kramers–Kronig “inconsistency” when we attempt to fit the unpolarized reflectance and transmittance spectra by isotropic dielectric function (see Supplementary Section 10 and Supplementary Figure 16f). In another paper², the authors find anisotropic optical properties by Mueller matrix ellipsometry. Yet, these constants are unable to explain our polarized transmission measurements (see Supplementary Figure 2a,c). By contrast, the dielectric tensor that we calculated from the first principles agrees significantly better. It reproduces the transmittance map on the qualitative level (see Supplementary Figure 2b) and gives an almost perfect match for the principal axis rotation angles, especially near 750 nm where the wandering principal axis phenomenon is the most pronounced. At the same time, we note some quantitative mismatch between the measured and calculated transmittance maps, which can be attributed to a slight inaccuracy of the first principle calculations for determination of the dielectric tensor.

Supplementary Figure 1: **a**, Optical constants of ReS_2 from paper¹ and **b**, Lorentz oscillator fitting of optical constants, presented in panel (a).

Supplementary Figure 2: **a**, Experimental transmittance for 150 nm-thick ReS_2 flake. **b**, Calculated by the transfer matrix method transmittance based on optical constants of ReS_2 from Figure 4. **c**, Calculated by the transfer matrix method transmittance based on optical constants of ReS_2 found from ellipsometry in the recent work².

Supplementary Section 8: Theoretical background of waveguide mode direction change – near field calculations analysis.

We consider a 150-nm-thick ReSe₂ layer on a glass substrate in our numerical near-field calculations. The refractive index of the glass is 1.4, whereas the ReSe₂ permittivity is described by the phenomenological 3-exciton model (see Supplementary Section 5 and Supplementary Figure 6 for details). We illuminate a 500-nm-radius hole in the layer by a plane wave normally incident from the substrate and study the z-component (perpendicular to the structure surface) of the electric field, $|E_z|$, in the air at a distance of 5 nm from the air/ReSe₂ interface. Due to the peculiar structure of the 3-exciton model permittivity tensor ($\hat{\epsilon}_{xz} = \hat{\epsilon}_{yz} = 0$) the normally incident plane wave does not induce a z-component in the background field, $E_{bg}^z = 0$. Therefore, the z component of the total electric field equals to the corresponding component of the scattered field $E_{tot}^z = E_{sc}^z$.

Supplementary Figure 7 presents a series of near-field distributions, categorized by the incident light wavelengths (columns) and angle of flake rotation (rows). The three columns represent wavelengths of 920 nm, 940 nm, and 950 nm, while the seven rows display fields for rotation angles of {0°, 30°, 60°, 90°, 120°, 150°, 180°}. Both 0° and 180° rotations yield identical results but are shown for the ease of comparison with adjacent panels. The incident light polarization (indicated by white arrows) is vertically aligned in all panels.

Upon comparing the columns, we observe that the slight rotation of the bright (yellow in the image) electrostatic-dipole-like-polarized field in the immediate vicinity of the hole is most apparent in the first and last rows. The waves surrounding the hole also exhibit slight shifts. Concurrently, the flake's rotation has a relatively weak effect on the bright, dipole-like field, which remains predominantly vertically polarized regardless of the flake's orientation. However, the waves surrounding the hole clearly rotate along with the flake. This supports the notion that these observed waves are associated with the waveguide modes of the anisotropic ReSe₂ layer.

Supplementary Figure 8: Near field distribution near a circular hole in the ReSe₂ layer. Three columns correspond to 920 nm, 940 nm, and 950 nm wavelengths. Eight rows correspond to six linear polarizations of incident light taken with a 30-degree step, left-handed (LCP), and right-handed (RCP) circular polarization. Near field distribution pattern in the considered anisotropic structure mainly depends on the polarization of the incident beam but also slightly varies with wavelength.

The numerical calculations not only provide the reader with the absolute value of the electric field but also its phase (see Supplementary Figure 8). To analyze them together, we employ a straightforward data processing algorithm, as illustrated for the case of a non-rotated flake and a wavelength of 920 nm in Supplementary Figure 9. The following steps are followed: *(i)* transformation of coordinates from (x, y) to (r, φ) (see Supplementary Figure 8); *(ii)* referencing^{1,8} and application of the Parzen window (see the window shape in Supplementary Figure 9) to smoothen the computational domain's edge and the hotspot near the hole (see Supplementary Figure 8); and finally, *(iii)* performing a Fourier transform on the obtained complex field for the radial coordinate and analyzing its absolute value (see Supplementary Figure 8). The application of the Parzen window enables us to extend the Fourier integral to infinite limits and avoid parasitic oscillations caused by the Gibbs phenomenon.

Supplementary Figure 8: Processing of the near-field distribution for a circular hole in the ReSe₂ layer. **a**, The absolute value and **b**, phase of the near-field distribution are shown for vertical polarization of 920 nm light incident on a non-rotated flake. **c-d**, The same fields after transformation to polar coordinates. **e-f**, Multiplication of the obtained field with a real-valued Parzen window does not affect the phase but smooths the absolute value of the field near the hole and edges of the computational domain. **g**, The absolute value of the obtained field Fourier image. The white dashed line indicates the wavevector of air modes, and the observed bright lines indicate the dispersion of air and waveguide modes.

Supplementary Figure 9: Shape of the Parzen filter applied to the near-field distributions.

The procedure described above allows us to effectively distinguish the contributions of different modes to the overall field. As depicted in Supplementary Figure 8, all the modes are situated in the upper half-space, indicating positive values of the radial wavevector, k_r . This confirms the evident fact that the modes propagate outward from the hole. One of the branches follows the horizontal white dashed line, which corresponds to the light wavevector in the air. This confirms that this particular mode is associated with freely propagating air modes scattered out by the hole. Moreover, the angle dependence of this mode's amplitude exhibits a typical vertical dipole radiation pattern - with dips observed at $\varphi_{dips} = \{0^\circ, 180^\circ\}$ and peaks in between. However, the peaks are significantly shifted from the expected positions of $\pm 90^\circ$, as the actual polarization currents induced by the hole scattering differ significantly from a point dipole that is vertically aligned with the polarization of the incident light. In addition to the air modes, we also observe branches of waveguide modes of two types. The first type exhibits strong angular dispersion and propagates in almost all possible directions, while the second type is observed in a narrow range of angles near $\varphi = \{0^\circ, 180^\circ\}$ and possesses much larger wavevectors.

Based on our findings regarding the mode structure, we conclude that the waves observed in Supplementary Figure 7 (a) (as well as in Supplementary Figure 8) are not associated with either air modes or waveguide modes individually, but rather with their interference: $|E_z(\varphi, r)| \propto |E_{air}(\varphi)e^{ik_0r} + E_{waveguide}(\varphi)e^{ik_{wg}r}|$. Therefore, the radial period of these waves is inversely proportional to the difference between their wavevectors, given by $\lambda_{waves} = \frac{2\pi}{k_{wg} - k_0} = \frac{\lambda_{wg}\lambda_0}{\lambda_0 - \lambda_{wg}} =$

$\frac{\lambda_0}{\beta_{wg}-1}$, where β_{wg} represents the index of the waveguide mode and λ_{waves} , λ_{wg} , λ_{wg} correspond to the period of the waves in the near-field, wavelengths of the waveguide and air modes, respectively.

By following the described procedure, we obtain the Fourier spectra (refer to Supplementary Figure 10) for various wavelengths and angles of flake rotation. All the observed effects are consistent with what we have already observed in real space (see Supplementary Figure 7), but they are much more straightforward. Notably, the transformation of the angular dispersion of the waveguide modes with wavelength becomes more apparent. The rotation of the flake results in a corresponding rotation of the waveguide modes, while the angular position of the "dipole-like" air mode remains mostly unaffected.

Supplementary Figure 10: Fourier spectra of the near-field distribution for a circular hole in the ReSe₂ layer. Each panel demonstrates the absolute value of the Fourier transform of the filtered near field, $|F[E_x(\varphi, r) \cdot f_{\text{parzen}}(r)]|(\varphi, k_r)$. The

three columns correspond to wavelengths of 920, 940, and 950 nm, while the even rows correspond to flake rotations of 0°, 30°, 60°, 90°, 120°, 150°, and 180°. The gray axes indicate the orientation of the flake, and the white vertical arrows show the polarization of the incident light. The white dashed horizontal lines correspond to the wavevector of air modes $k_0 = \pm \frac{2\pi}{\lambda_0}$.

Supplementary Section 9: Comparison of calculated and measured near-fields.

In the previous section, we discussed the contribution of air modes and waveguide modes to the near field and explained the observed waves through their interaction. To analyze this interaction, we utilized both the absolute value and phase of the electric field. However, due to limitations in experimental measurements, we are unable to obtain a reliable phase of the measured signal. As a result, direct comparison between the Fourier spectra of calculated fields (see Supplementary Figure 10) and processed experimental data is not possible. To address this issue, we applied the same postprocessing procedure to the calculated fields, considering only their absolute values and disregarding the phase (see Supplementary Figure 11). Since the absolute value of an electric field is a real-valued function, its Fourier image is an even function of the wavevector (see Supplementary Figure 11). The absolute value of the near-field does not differentiate between the contributions of air and waveguide modes. Consequently, in the spectra, we only observe the lines corresponding to the waves observed in the near field, which result from the interference of these modes. As mentioned earlier, the wavevector of these modes is determined by the difference $|k_{waves}| = k_{wg} - k_0$. In other words, waveguide modes are "shifted" by a wavevector k_0 towards the origin, while still exhibiting angular dispersion similar to the original waveguide modes. Additionally, the bright horizontal line at $k_r = 0$ is associated with the "mean background" field, which is determined by the amplitudes of both air and waveguide modes, as well as other non-resonant contributions of the scattered field.

Supplementary Figure 12: Processing of the near-field distribution for a circular hole in the ReSe₂ layer. **a**, The absolute value of the near-field distribution is shown for vertical polarization of 920 nm light incident on a non-rotated flake. **b**, The same fields after transformation to polar coordinates. **c**, Multiplication of the obtained field with a real-valued Parzen window smooths the absolute value of the field near the hole and edges of the computational domain. **d**, The absolute value of the obtained field Fourier image. The white dashed lines indicate the wavevector of air modes, and the observed bright lines are associated with the mean background field ($k_x = 0$) and interference of waveguide and air modes.

We repeated the procedure for different values of wavelength and flake orientations, as shown in Supplementary Figure 13. Although the Fourier spectra of the absolute value of the near field appear slightly different from the previously demonstrated ones, all the described effects are still present. The waveguide modes still rotate with the flake, while the background-associated harmonics are more stable.

Supplementary Figure 13: Fourier spectra of the near-field distribution for a circular hole in the ReSe₂ layer. Each panel demonstrates the absolute value of the Fourier transform of the filtered near-field absolute value, $|F[E_z(\varphi, r) \cdot f_{\text{Parzen}}(r)](\varphi, k_r)|$. The three columns correspond to wavelengths of 920, 940, and 950 nm, while the even rows correspond to flake rotations of 0°, 30°, 60°, 90°, 120°, 150°, and 180°. The gray axes indicate the orientation of the flake, and the white vertical arrows show the polarization of the incident light. The white dashed horizontal lines correspond to the wavevector of air modes $k_0 = \pm \frac{2\pi}{\lambda_0}$.

A comparison between the numerical estimations and experimental results can be found in Supplementary Figure 14. Our calculations reproduce the main experimental peculiarities, such as suppressed scattering in the first and third quadrants and a slight increase in the period of the waves with angle in the upper and lower semispaces.

Supplementary Figure 14: Comparison of the numerically calculated and experimentally measured near-field distributions for a circular hole in the ReSe₂ layer. The comparison is shown for wavelengths of 920, 940, and 950 nm of incident light for both real-space distribution and corresponding Fourier spectra. A photograph demonstrates the orientation of the flake in the microscope.

COMMENT 4: The authors write that they visualize “asymmetry of the waveguide modes (Figures 3b-d) caused by material anisotropy only.” Yet, waveguide modes with weak confinement are known to have angular dependence in SNOM measurements (see eg. Ref. [29]). The authors are suggested to extract the SNOM wavevectors from their data as a function of the in-plane angle, perform established angular correction protocols (described, for instance in Ref. [34]) and compare the results with calculations using their reported optical constants.

RESPONSE: We thank the Reviewer for the comment and agree that this effect occurs in reflection-mode measurements and that it can be difficult to distinguish the effect of rotation of optical constants from this effect. This is the reason why we decided to perform the main measurements in transmission-mode, where there is no such effect. Nevertheless, we agree with the Reviewer that this matter needs further clarification, so we made additional near-field measurements in reflection-mode and modified the revised version of our manuscript by:

As a practical demonstration, we show the effect of wavelength–dispersive principal optical axes on waveguide mode propagation direction using a scattering scanning near-field optical microscopy (s-SNOM) in the transmittance scheme, depicted in Figure 3a, since the reflectance scheme has angular rotation (see Supplementary Section 6).

We also modified Supplementary Information by:

Supplementary Section 6: Angular dependence of weak confinement modes in the reflection scheme of near-field microscopy.

Supplementary Figure 7 shows near-field measurements of the ReSe₂ disk with a thickness of 125 nm in reflection mode. The strong angular dependence, which can be seen in Supplementary Figure 7 is caused by the geometry of the experiment. This effect is related to momentum conservation along the edge direction^{1,6}. And can be described as:

$$n_{eff} = n_{SNOM} + \cos(\gamma) \sin(\beta),$$

where γ is the angle between the illumination wavevector k and its projection k_{\parallel} on the sample surface plane and β is the angle between k_{\parallel} and the sample edge. Thus, the sinusoidal dependence shown in the Supplementary Figure 7 is associated with the rotation of the edge of the sample. Due to the described effect, it is difficult to observe the effect of rotation of the optical axes; therefore, the measurements in the main text were carried out in the transmission mode, where there is no such angular dependence.

Supplementary Figure 7: a, Scanning electron microscopy image of the sample. b, atomic force microscopy image of the measured ReSe₂ disk. c-d, Amplitude and phase of the near-field signal measured in reflection-mode. e, Unwrapped, relative to the center, measurements from c and d, green points on e indicate profile dependence of the mode on the in-plane angle of rotation. f, Comparison of the mode's profile picked from e and sinusoidal dependence on the angle α .

COMMENT 5: I found the opening sentence of the abstract “Nature is abundant in material systems...” confusing. The authors are suggested to rephrase.

RESPONSE: We thank the Reviewer for the comment. We removed the word “systems” from the abstract in the revised version of our manuscript:

Nature is abundant in material platforms¹ with anisotropic permittivities arising from symmetry reduction that feature a variety of extraordinary optical effects, such as negative refraction², topological optical transitions³, and light canalization⁴.

COMMENT 6: The authors cite ref [26] for negative refraction of polaritons on pg. 2. However, while Ref. [26] demonstrated so-called ‘bending-free’ refraction, negative refraction was not claimed in this report. Negative refraction of polaritons has been reported in H. Hu et al., Science 379, 558 (2023), and A. J. Sternbach et al., Science 379, 555 (2023)

RESPONSE: We are grateful to the Reviewer for the comment. We updated the reference list in the revised version of our manuscript:

This anisotropy produces complex isofrequency contours in the reciprocal space²⁴ enabling hyperbolic materials²¹, ghost¹ and shear^{10, 25} polaritons, negative refraction^{26, 27}, canalization of radiation⁴, and many other intriguing wave phenomena.

26. Hu, H. *et al.* Gate-tunable negative refraction of mid-infrared polaritons. *Science*. **379**, 558–561 (2023).

27. Sternbach, A. J. *et al.* Negative refraction in hyperbolic hetero-bicrystals. *Science*. **379**, 555–557 (2023).

COMMENT 7: The authors mention a “super-prism effect” on page 6 but no reference is provided. The authors are suggested to include a reference.

RESPONSE: We thank the Reviewer for the careful reading. We added the reference for the super-prism effect in the revised version of our manuscript.

Moreover, the non-straight orientation of the principal optical axes of the permittivity tensor leads to slanted isofrequency surfaces with respect to the global z-axis (Supplementary Figure 11), which may enable interesting transmission phenomena, such as negative refraction and the super-prism effect⁵⁴.

54. Kosaka, H. *et al.* Superprism phenomena in photonic crystals. *Phys. Rev. B* **58**, R10096–R10099 (1998).

REVIEWER COMMENTS

Reviewer #2 (Remarks to the Author):

The authors have addressed my questions well and I'd like to recommend it.

Reviewer #3 (Remarks to the Author):

The authors have provided detailed responses to all concerns. However, several strong concerns with the current presentation remain. If the authors can respond to these concerns adequately, I believe a quick decision can be made on the manuscript:

1) "Phenomenological dielectric permittivity" is highly confusing. If one axis of this quantity is negative while others remain positive is the crystal hyperbolic?

If the plot is misleading and does not describe the actual permittivity of ReS₂ or explain experimental observables the authors are strongly advised to remove this plot. It would be interesting to see how the results of the "two exciton model" in fig 2h change with the linewidth along yy and xy directions.

2) The analysis in Fig.S8 appears to be perfectly applied. However, the analysis in Fig. S12-14 remains confusing. In particular, the authors claim that the absolute value of the wavevector is obscured by a complex interference term that they were unable to resolve.

First the presentation does not agree with the text. The text claims the feature of the air mode is pushed to $k=0$ in Fig.S12-14 while the figure still shows a white dashed line indicating the air mode at finite momentum. This gives the appearance, in Fig.14 that the primary feature observed in the data at finite momentum appears at a wavevector nearly identical to the wavevector of the air mode.

Second, if the authors cannot resolve the absolute value of the wavevector due to lack of phase data, the question remains: are the authors confident that they are observing a waveguide mode and not only an air mode?

3) To the best of my knowledge, angular dependence does not uniquely come from a reflective geometry. The physical origin is light scattered from the tip has a difference of phase from radiation scattered from a remote object (or edge) at a distance of x from the tip. The phase accumulation depends on the angle. Section S6 does not clarify why this cannot happen in a transmission geometry.

If the authors would like to refute the possibility of angular dependence in the transmission geometry, data similar to that shown in Fig. S7 should be obtained in a transmission geometry and analyzed for angular dependence.

**SUMMARY OF THE CHANGES INTRODUCED IN MANUSCRIPT NCOMMS-
23-26804-T**

**"WANDERING PRINCIPAL OPTICAL AXES IN VAN DER WAALS
TRICLINIC MATERIALS"**

Our detailed responses to all Reviewers' comments are listed below, and additionally, are marked by green in the revised version of our manuscript.

RESPONSES TO REVIEWER #2's COMMENTS AND CONCERNS:

GENERAL ASSESSMENT: The authors have addressed my questions well and I'd like to recommend it.

RESPONSE: We thank the Reviewer for positive evaluation of our work and recommendation of our manuscript.

RESPONSES TO REVIEWER #3's COMMENTS AND CONCERNS:

GENERAL ASSESSMENT: The authors have provided detailed responses to all concerns. However, several strong concerns with the current presentation remain. If the authors can respond to these concerns adequately, I believe a quick decision can be made on the manuscript:

RESPONSE: We thank the Reviewer for valuable comments and careful assessment. We hope that the revised version of our manuscript clarifies all the raised questions and comments.

COMMENT 1: "Phenomenological dielectric permittivity" is highly confusing. If one axis of this quantity is negative while others remain positive is the crystal hyperbolic?

If the plot is misleading and does not describe the actual permittivity of ReS₂ or explain experimental observables the authors are strongly advised to remove this plot. It would be interesting to see how the results of the "two exciton model" in fig 2h change with the linewidth along yy and xy directions.

RESPONSE: We are grateful to the Reviewer for the comment. Our initial goal for the bi-exciton model was to provide a phenomenological explanation of wandering principal optical axes' origin. At the same time, we agree with the Reviewer that it our model would be much more valuable if, in addition to orientation of optical axis, it also describes the optical response of ReS₂. For this purpose, in the revised version of our manuscript, we started our fitting procedure taking optical constants from a recent work [Mooshammer, F. et al. In-Plane Anisotropy in Biaxial ReS₂ Crystals Probed by Nano-Optical Imaging of Waveguide Modes. ACS Photonics 9, 443–451 (2022)]. Figure 2g shows the resulting bi-exciton model's dielectric permittivity, which is now in a close agreement with the dielectric responses reported in previous works [Mooshammer, F. et al. In-Plane Anisotropy in Biaxial ReS₂ Crystals Probed by Nano-Optical Imaging of Waveguide Modes. ACS Photonics 9, 443–451 (2022) and Munkhbat, B., Wróbel, P., Antosiewicz, T. J. & Shegai, T. O. Optical Constants of Several Multilayer Transition Metal Dichalcogenides Measured by Spectroscopic Ellipsometry in the 300–1700 nm Range: High Index, Anisotropy, and Hyperbolicity. ACS Photonics 9, 2398–2407 (2022)] and correctly describes wandering optical axes (Figure 2h).

Figure 2 | Observation of wandering (wavelength-dispersive) principal optical axes in triclinic ReS₂. **a**, Optical and **b**, ellipsometry micrographs of bulk ReS₂. **c**, Polarized transmittance of bulk ReS₂ presented in panel (a), for three different excitons at 830 nm (exc-1), 816 nm (exc-2), and 774 nm (exc-3). Polarized transmittance **d**, spectra and **e**, contour map. Red and blue points show the positions of in-plane principal optical axes. Dashed lines correspond to the crystallographic *a*-axis (red line) and *b*-axis (blue line). Zero degree corresponds to the crystallographic *b*-axis. The red and blue points are obtained through the fitting of

polarization-resolved microtransmittance at each wavelength (see Methods section Determination of principal optical axes). **f**, Depiction of non-orthogonal excitons (phenomenological theory). **g**, Dielectric tensor corresponding to the *bi*-exciton model. **h**, Principal optical axes orientation as a function of wavelength.

COMMENT 2: The analysis in Fig.S8 appears to be perfectly applied. However, the analysis in Fig. S12-14 remains confusing. In particular, the authors claim that the absolute value of the wavevector is obscured by a complex interference term that they were unable to resolve.

First the presentation does not agree with the text. The text claims the feature of the air mode is pushed to $k=0$ in Fig.S12-14 while the figure still shows a white dashed line indicating the air mode at finite momentum. This gives the appearance, in Fig.14 that the primary feature observed in the data at finite momentum appears at a wavevector nearly identical to the wavevector of the air mode.

Second, if the authors cannot resolve the absolute value of the wavevector due to lack of phase data, the question remains: are the authors confident that they are observing a waveguide mode and not only an air mode?

RESPONSE: We are grateful to the Reviewer for thorough analysis of our results and helpful feedback.

1) We acknowledge the confusion with the dashed lines in the graphs. Initially, they were introduced in Fig.S9 and Fig.S11 to represent the air mode. However, they were carried over to Fig.S12-14 as a convenient reference mark for all the graphs. Nevertheless, these dashed lines are not directly related to the lines observed in Fig. S12-14. To avoid any misinterpretation, we have removed the dashed lines from Fig.S12-14 and corresponding captions in the revised version of our manuscript. Additionally, we have changed the colormaps to "hot" colorscale to improve the visibility and the consistency with the ones in the main text.

2) Indeed, the lack of experimentally measured phase complicates the analysis significantly. However, we are confident that the observed modes are not just air modes. We have several arguments to support this claim.

a) As mentioned, the dashed lines in Fig.S14 should not be associated with the experimentally observed lines. Nevertheless, they serve as a reference for comparison with the air modes. For instance, below we show the Fourier spectrum of experimental measurements at a wavelength of 940 nm, with qualitatively indicated angular dependence of the observed mode (dashed light-blue lines). While the changes in k -vector may not be very strong, they are still clearly visible. In contrast, the wavevector of the air mode (indicated by white dashed lines) should be equal to $k_0 = 2\pi/\lambda$ for all the directions. These air modes can be seen in Supplementary Figure 12 and behave accordingly. .

b) Additionally, the k -vector of the experimentally observed mode is slightly smaller than k_0 for certain propagation directions (indicated by green ellipses below). In other words, the period of oscillations in the experimentally measured near field is slightly larger than a wavelength of air mode. Furthermore, other potential modes, such as modes from bulk ReSe₂, substrate and waveguide modes would have even smaller wavelengths than the air modes and cannot explain the observed effect. The typical mechanism for the formation of larger period oscillations from smaller period ones is interference, which is discussed in our study. Therefore, we believe that our interpretation of the observed mode as an interference of waveguide and air modes is accurate.

COMMENT 3: To the best of my knowledge, angular dependance does not uniquely come from a reflective geometry. The physical origin is light scattered from the tip has a difference of phase from radiation scattered from a remote object (or edge) at a distance of x from the tip. The phase

accumulation depends on the angle. Section S6 does not clarify why this cannot happen in a transmission geometry.

If the authors would like to refute the possibility of angular dependence in the transmission geometry, data similar to that shown in Fig. S7 should be obtained in a transmission geometry and analyzed for angular dependence

RESPONSE: We thank the Reviewer for the clarification question. Indeed, angular dependency also can appear in a transmission geometry. However, we employed an option of synchronization between incident light and irradiated region (in our case, hole), designed earlier by SDU team together with the Neaspec team [V.A. Zenin et al., “Boosting Local Field Enhancement by on-Chip Nanofocusing and Impedance-Matched Plasmonic Antennas”, *Nano Letters*, 15(12), 8148 – 8154 (2015)]. In this configuration, the angular dependence can be filtered out, as explained in the modified version of our manuscript:

This effect is associated with a different phase shift of the light scattered by the tip of the probe and the distant object (edge of the disk). Light focused on the tip of the probe is converted into evanescent waves, which, upon reaching the sample-air interface, are divided into two beams: one part is elastically reflected from the surface then scattered again on the probe and returned to the detector, the other part excites cylindrical waveguide modes in the sample, which scatter upon reaching the edge of the sample (see Supplementary Figure 8a). Since a parabolic mirror collects the light at an angle relative to the normal of the sample, an additional phase shift occurs between these two parts of the light due to different propagation lengths in the air. The phase shift depends on the distance x from the edge of the sample to the tip, thus, the interference between those two beams changes the observed oscillation frequency of the waveguide mode. The effect can be expressed as:

$$n_{eff} = n_{SNOM} + \cos(\gamma) \sin(\beta)$$

where γ is the angle between the illumination wavevector k and its projection k_{\parallel} on the sample surface plane, and β is the angle between k_{\parallel} and the sample edge. Thus, the sinusoidal dependence shown in Supplementary Figure 7 is due to the change of the position of the sample edge on which scattering occurs. It is important to note that in this case, both parts of the scattered light (from the probe and from the edge) behave as near-field: they quickly decay as the probe moves away from the surface. In the case of transmittance measurements (see Methods in the main text), we illuminate the sample along the normal from below focusing light on the hole. Using synchronized movement of the sample and the lower parabolic mirror relative to the probe and the upper parabolic mirror⁷, we can ensure that the light always remains focused at the hole, and does not move with the probe, as in the case of reflection. When light is scattered by a hole, one part of the scattered radiation goes directly to the detector, the other part couples into waveguide modes, and after passing through the sample, is scattered by the probe, as seen in Supplementary Figure 8b. It is important to note that although such a system is very similar to the system described above, the interference effect responsible for the spatial frequency shift will not occur

since the radiation directly scattered by the hole to the detector does not behave like near-field radiation and does not diminish when the tip of the probe moves away from the sample. Therefore, it will be filtered out by a heterodyne system⁸ and will not be observed in the background-free images (harmonics ≥ 3). However, in images that are not completely free from background (harmonic = 1), the radiation that comes directly from the hole is not suppressed completely, and it is possible to observe frequency shift effect similar to that in the reflection mode (see Supplementary Figure 8c-d). It is important to point out that even in the image with significant background it is still possible to observe angular dependence related to the anisotropy of the material ($\sim \sin(2\alpha)$), however according to our calculations this effect is more than 2.5 times weaker than spatial frequency shift related to the interference with background ($\sim \sin(\alpha)$). Accordingly, in the background-free transmittance images, we do not observe angular dependence of $\sim \sin(\alpha)$, and instead, we observe $\sim \sin(2\alpha)$ dependence, which comes from the in-plane anisotropy of the material. The absence of the spatial frequency shift in the background-free transmittance measurements is one of the main reasons why we carried out our main measurements using this technique.

Supplementary Figure 7: **a**, Scanning electron microscopy image of the sample. **b**, atomic force microscopy image of the measured ReSe_2 disk. **c-d**, amplitude and phase of the near-field signal measured in the reflection mode. **e**, unwrapped, relative to the center, measurements from **c** and **d**, the green points in panel **e** indicate profile dependance of the mode on the in-plane angle of rotation. **f**, comparison of the mode profile picked from panel **e** and sinusoidal dependence on the angle α .

Supplementary Figure 8: a-b, Reflection and transmittance schemes in near-field microscopy. c-d, near-field image with first and third harmonics filtration. e-f, unwrapped, relative to the center, measurements from c and d, the blue points indicate the dependance of the mode profile on the in-plane angle of rotation, the red curve represents a sinusoidal fit with arguments α and 2α , respectively.

REVIEWERS' COMMENTS

Reviewer #3 (Remarks to the Author):

The authors have addressed all concerns and I can now recommended their manuscript for publication.